# PromptDyG: Test-Time Prompt Adaptation on Dynamic Graphs

**Guoguo Ai** [1 2 *]  **Chaoxi Niu** [3 *]  **Hui Yan** [1]  **Joey Tianyi Zhou** [4 5]  **Yew-Soon Ong** [6]  **Guansong Pang** [2 †]

## Abstract

Activities in numerous evolving systems can be represented as dynamic graphs in snapshot form at different time intervals, *i.e.*, discrete-time dynamic graphs (DTDGs). Existing methods show impressive advances in capturing historical temporal evolution patterns in DTDGs, but they focus on addressing an offline learning setting, where models are trained using historical snapshots once and then evaluated to all subsequent graph snapshots without further updating. This fails to capture **1)** the nature of evolving complexities across graph snapshots and **2)** the distribution shift in the testing graph snapshots. To address these problems, we propose **PromptDyG**, a novel framework that leverages unsupervised test-time Prompt adaptation for Dynamic Graph learning under a live-update online setting. The key insight is that an expressive dynamic graph prompt can be learned on a frozen backbone via minimization of feature-wise, label-free entropy to efficiently and continuously model the evolving patterns. We show theoretically that this unsupervised prompt adaptation can guarantee a larger similarity margin between positive and negative pairs, facilitating more accurate dynamic predictions. It is further confirmed by our extensive empirical results on six benchmark datasets that show consistent and significant improvements of PromptDyG over state-of-the-art baselines. Code is available at https://github.com/mala-lab/PromptDyG.

## 1. Introduction

Due to the superior capacity to represent complex relations between samples, graph is widely used in various real-world applications (Xia et al., 2021; Zhang et al., 2020; Ai et al., 2025), such as social networks, bank transaction networks, and online shopping. For example, in the context of bank transaction networks, there are dynamic intersections among customers where nodes and edges represent customers and their transactions, respectively. In real-world applications, such a dynamic nature means that the number of nodes and the connectivity between nodes would change over time (Pareja et al., 2020; Kazemi et al., 2020). This dynamicity poses great challenges for traditional Graph Neural Networks (GNNs), which focus on static graphs with fixed nodes and edges. To accommodate temporal evolution, various GNN-based models are proposed for dynamic graph learning (Manessi et al., 2020; Pareja et al., 2020; Zhao et al., 2019; Seo et al., 2018; You et al., 2022; Zhu et al., 2023; Qi et al., 2025).

Dynamic graphs can be modeled at two main temporal granularities, including Continuous-time Dynamic Graphs (CT-DGs) and Discrete-time Dynamic Graphs (DTDGs). CT-DGs (Kumar et al., 2019; Cong et al., 2023; Yu et al., 2024) represent graph evolution as a stream of fine-grained temporal events. In contrast, DTDGs (Yang et al., 2021; You et al., 2022; Qi et al., 2025) describe graph evolution as a sequence of snapshots aggregated over different time intervals, such as days or weeks. In this paper, we focus on DTDGs as the snapshot-based formulation is particularly important in many real-world applications, where interactions are naturally collected in batches, since precise timestamps may be unavailable and/or system-level patterns within a time window are of primary interest.

Existing studies for DTDGs focus on learning representations on separate snapshots using static GNNs (Kipf & Welling, 2017; Veličković et al., 2018). Then, a temporal module is incorporated into the static GNNs to capture historical temporal evolution patterns in DTDGs for predicting future snapshots. However, these DTDG models (Zhao et al., 2019; Pareja et al., 2020; Qi et al., 2025) typically adopt an offline learning setting, as shown in Figure 1 (b), where models are trained using historical snapshots once and then evaluated for all subsequent snapshots without further up-

---

[*]Equal contribution  [1]School of Computer Science and Engineering, Nanjing University of Science and Technology, China [2]School of Computing and Information Systems, Singapore Management University, Singapore [3]Faculty of Data Science, City University of Macau, Macau, China [4]Centre for Frontier AI Research (CFAR), Agency for Science, Technology and Research (A*STAR), Singapore [5]Institute of High Performance Computing (IHPC), Agency for Science, Technology and Research (A*STAR), Singapore [6]College of Computing and Data Science, Nanyang Technological University, Singapore. Correspondence to: Guansong Pang <gspang@smu.edu.sg>.

*Proceedings of the 43rd International Conference on Machine Learning*, Seoul, South Korea. PMLR 306, 2026. Copyright 2026 by the author(s).

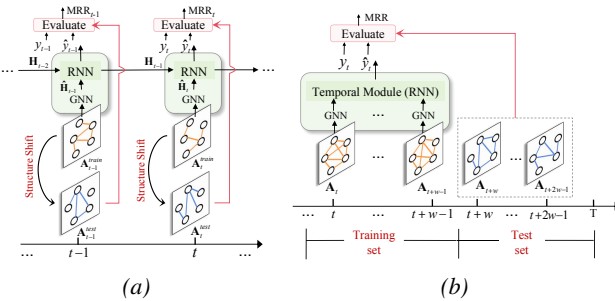

*(a)*             *(b)*

*Figure 1.* (a) *Online setting:* the model updates continuously to adapt to evolving data. (b) *Offline setting:* the first portion of snapshots is used for training, with the rest for evaluation, under a moving window of size $w$.

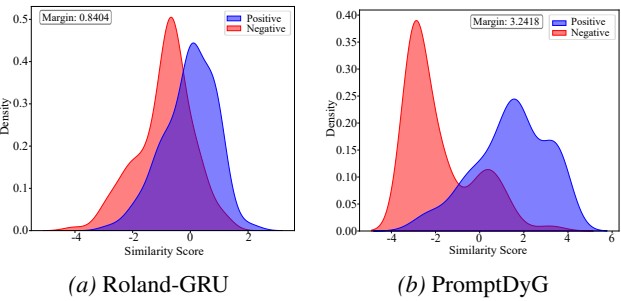

*(a)* Roland-GRU       *(b)* PromptDyG

*Figure 2.* Similarity score distributions for positive and negative pairs from (a) the backbone model Roland-GRU (You et al., 2022) and (b) PromptDyG at the same test snapshot in the dataset UCI (Panzarasa et al., 2009). The margin is defined as the difference between the mean similarity scores of positive and negative pairs.

dating. This setting fails to consider the nature of evolving complexities across graph snapshots, where each snapshot is generated chronologically and its distribution differs from those across testing graph snapshots. Consequently, the learned models are not adapted to those potential distribution shift, resulting in inferior performance.

To address these problems, we propose a novel framework that leverages unsupervised test-time Prompt adaptation for Dynamic Graph learning (**PromptDyG**) under a live-update online setting (You et al., 2022). As shown in Figure 1 (a), this online setting requires a dynamic model to predict future events based on previously observed snapshots and then incorporate the knowledge of new graph snapshots to fit evolving data. One key challenge is that, even within the same graph snapshot, the distribution of the training and test data can vary significantly due to the dynamic nature, inducing large structural distribution shifts. Current state-of-the-art (SOTA) methods (You et al., 2022; Chen et al., 2024; Qi et al., 2025) such as Roland (You et al., 2022) neglect this shift, which can lead to large confusion between positive (existing edges) and negative (non-existing edges) node pairs, as shown in Figure 2 (a).

To mitigate the distribution shift, graph test-time adaptation (TTA) (Wang et al., 2022; Chen et al., 2022b; Bao et al., 2025; Jin et al., 2023) has emerged as a promising approach by fine-tuning the pre-trained model with the test graph from different domains or performing test graph augmentation. While these methods achieve promising results for static graphs, they are not applicable to DTDGs. First, these methods either undermine the temporal dynamics encoded in the pre-trained DTDG models or introduce significant computational overhead. Second, DTDGs aim to predict the subsequent graph snapshots based on the observed data, which is fundamentally different from the test graph setup in non-dynamic graph test-time adaptation. To address these problems, the proposed PromptDyG introduces an unsupervised prompt adaptation stage to bridge the distributional gap. The key insight is that a lightweight dynamic graph prompt can be learned on a frozen backbone via minimization of feature-wise, label-free entropy to efficiently and continuously model the evolving patterns. Specifically, at each time step, the snapshot-specific pre-trained model remains frozen, and a lightweight learnable prompt is incorporated into the feature space of the test graph. This helps avoid distorting the time-related graph structures and preserves the integrity of the pre-trained model. The prompt is then optimized via unsupervised entropy minimization to align the representations with dominant evolutionary patterns under test structural shifts. This helps enlarge the similarity margin between positive and negative pairs (Figure 2(b)).

The main contributions are summarized as follows.

- We reveal that the distribution shift between training and test data hinders the effectiveness of existing DTDG methods on dynamic graphs.

- To address this issue, we propose **PromptDyG**, a simple yet theoretically sound unsupervised test-time prompt adaptation via feature-wise and label-free entropy minimization on DTDGs under structural shifts.

- Extensive experiments on six datasets demonstrate that PromptDyG significantly outperforms SOTA DTDG and graph TTA methods and can also consistently enhance diverse DTDG methods as a plugin module.

## 2. Related Work

**Dynamic GNNs on DTDGs.** Discrete-time dynamic graph representation learning typically combines spatial models (e.g., GCN (Kipf & Welling, 2017), GAT (Veličković et al., 2018)) to encode snapshots with temporal models (e.g., LSTM (Hochreiter & Schmidhuber, 1997), GRU (Dey & Salem, 2017)) to capture temporal dynamics. Early works such as WD-GCN (Manessi et al., 2020), EvolveGCN (Pareja et al., 2020), TGCN (Zhao et al., 2019) and GCRN

(Seo et al., 2018) adopt "GNN+RNN" architectures to model spatial and temporal information jointly. Several subsequent works have explored alternative designs for better efficiency and scalability. For instance, WinGNN (Zhu et al., 2023) adopts meta-learning strategies, and SFDyG (Qi et al., 2025) fuses multiple snapshots into a single temporal graph. While significantly advancing spatio-temporal modeling, they adopt the offline learning setting, which cannot fully reflect the evolving nature of dynamic graphs. To achieve online learning, Roland proposes a live-update protocol to mimic real-world application scenarios. However, like other DTDG models, Roland (You et al., 2022) focuses on capturing the evolution patterns of historical data, overlooking the test-time distribution shifts. To this end, we propose a test-time prompt adaptation framework to tackle the performance degradation caused by test-time structural shifts. See Appendix C for more details on dynamic GNNs.

**Test-Time Adaptation.** Test-Time Adaptation (TTA) has emerged as a promising paradigm for adapting models to shifting test distributions without re-accessing training data (Wang et al., 2021a; Chen et al., 2022a; Karmanov et al., 2024; Zhao et al., 2025). While early TTA focused on image-based data by entropy minimization (Wang et al., 2021a; Zhao et al., 2023; Lim et al., 2023), pseudo-labeling (Iwasawa & Matsuo, 2021; Zhang et al., 2023), or consistency regularization (Boudiaf et al., 2022), recent efforts have extended TTA to graph-structured data, falling into two categories, *i.e.*, test-time model adaptation and test-time graph adaptation. The former updates pre-trained GNN parameters via self-supervised objectives, *e.g.*, GT3 (Wang et al., 2022), GraphTTA (Chen et al., 2022b), and Matcha (Bao et al., 2025). In contrast, test-time graph adaptation takes a data-centric approach to modify the test graph data while keeping the pre-trained GNN model parameters unchanged. Typically, GTrans (Jin et al., 2023) performs test-time adaptation by augmenting the target graph and optimizing the contrastive objective. Nevertheless, augmentation-based approaches inevitably introduce high additional computational costs (Bao et al., 2025). Despite those TTA methods having proven to be effective in tackling distribution shift for static node and graph classification tasks, their application remains largely under-explored for dynamic graph learning.

**Graph Prompting Learning.** Prompt learning seeks to adapt pre-trained models to downstream tasks by incorporating learnable prompts while keeping the pre-trained models frozen (Liu et al., 2023a; Sun et al., 2023b). Specifically, it designs task-specific prompts capturing the knowledge of the corresponding tasks and enhances the compatibility between inputs and pre-trained models. Recently, prompt learning has been explored in graphs to unify multiple graph tasks (Sun et al., 2023a; Liu et al., 2023b; Fang et al., 2024) or improve the transferability of graph models on the datasets across domains (Li et al., 2024; Zhao et al.,

2024; Niu et al., 2024a;b; Qiao et al., 2025). Some methods (Yu et al., 2024; Yang et al., 2024) also incorporate graph prompting into dynamic graph learning, but they primarily focus on offline learning. This paper aims to propose an unsupervised test-time prompt adaptation to address the distribution shifts during inference under the online setting.

## 3. Preliminaries

**Discrete-Time Dynamic Graphs.** A static graph is generally defined by its nodes and edges, denoted as $G = (V, E)$, where $V = \{v_1, v_2, \cdots, v_N\}$ is the set of nodes, $E \subseteq V \times V$ is the set of edges. The structure of the graph is typically represented as an adjacency matrix $\mathbf{A} \in \{0, 1\}^{|V| \times |V|}$, and node features are denoted as a feature matrix $\mathbf{X} \in \mathbb{R}^{|V| \times D}$, where $D$ is the feature dimension, and $|V| = N$ is the number of nodes. Snapshot-based DTDGs extend static graphs by incorporating a temporal dimension. Specifically, each node $v$ is associated with a timestamp $\tau_v$ and each edge $e$ is associated with a timestamp $\tau_e$. At time step $t$, the graph snapshot is represented as $G_t = (V_t, E_t)$, where $V_t = \{v \in V | \tau_v = t\}$ and $E_t = \{e \in E | \tau_e = t\}$ denote the sets of nodes and edges at time $t$, respectively. The dynamic graph is therefore represented as a sequence of graph snapshots over $T$ time steps, denoted as $\mathcal{G} = \{G_t\}|_{t=1}^T$. We use $\mathbf{A}_t$ to denote the binary adjacency matrix corresponding to the edge set $E_t$.

**Representation Learning of DTDGs.** In the live-update setting, the edge set $E_t$ at each time step $t$ is partitioned into disjoint sets, *i.e.*, $E_t = E_t^{train} \cup E_t^{val} \cup E_t^{test}$, which are used to construct the corresponding adjacency matrices for training ($\mathbf{A}_t^{train}$), validation ($\mathbf{A}_t^{val}$), and test ($\mathbf{A}_t^{test}$). Then, at each time step $t$, a dynamic graph representation learning model, denoted as $f_\theta^t(\cdot)$, maps the current graph information and historical temporal representation $\mathbf{H}_{t-1}$ into a latent embedding space:

$$\mathbf{H}_t = f_\theta^t(\mathbf{A}_t^{train}, \mathbf{X}, \mathbf{H}_{t-1}), \tag{1}$$

where $\mathbf{H}_t$ represents the learned node embeddings at time $t$, which can be used for different tasks, such as link prediction.

## 4. Methodology

Existing DTDG models could effectively capture spatio-temporal evolution by integrating GNNs and RNNs. However, the test graph snapshots would exhibit distinct topological characteristics from those in the training data. As a result, the pre-trained model may produce pattern-blurred embeddings, resulting in an ambiguous distinction between positive and negative node pairs of the next snapshot. To this end, this paper proposes a novel test-time prompt adaptation framework for DTDGs, termed PromptDyG. As shown in Figure 3, with the backbone frozen, PromptDyG introduces

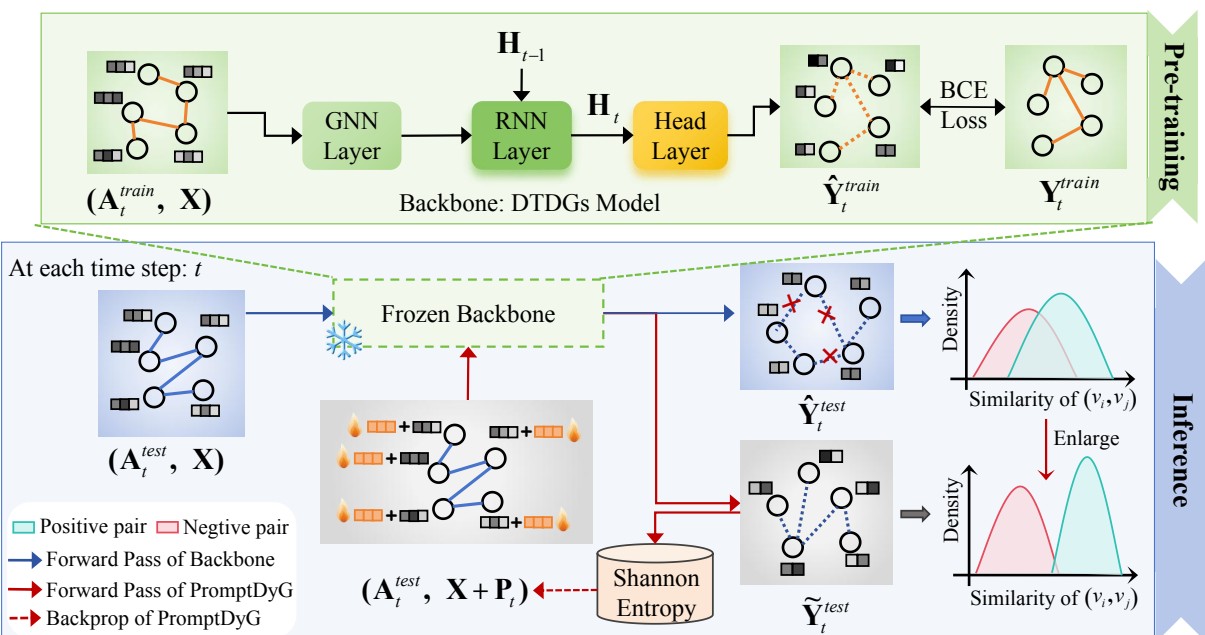

*Figure 3.* Overview of PromptDyG at time step $t$. In the pre-training phase (top), a standard DTDG backbone is trained via supervised Binary Cross-Entropy (BCE) loss to capture spatio-temporal dependencies for future link prediction. In the inference phase (bottom), a learnable prompt $\mathbf{P}_t$ is optimized via minimizing feature-wise, label-free entropy with the backbone parameters frozen. This process adapts the representation to the structural shift, resulting in distinct similarity distributions for positive and negative pairs.

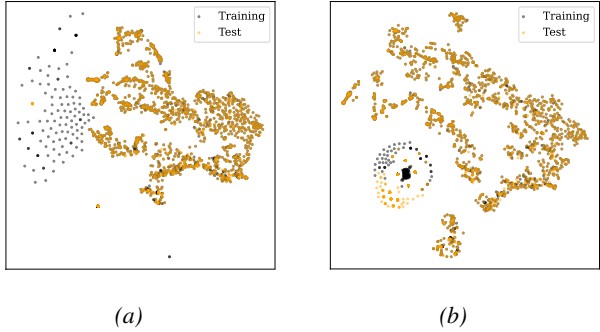

*Figure 4.* t-SNE visualization of training and test-time embeddings on UCI at the same time step. (a) Backbone; (b) PromptDyG.

an unsupervised feature-wise entropy minimization objective for tuning the learnable prompt $\mathbf{P}_t$ to refine the evolving patterns. Both theoretical analysis and empirical evaluations demonstrate that our method effectively enhances the model's generalization capability by enlarging the similarity margin between positive and negative pairs.

### 4.1. Test-Time Prompting on Dynamic Graphs

**Pre-training for Temporal Link Prediction.** For link prediction of next time step, a prediction head $g(\cdot)$ is used to compute the similarity score $s_{uv}$ between a pair of nodes $(u, v)$, *i.e.*, $s_{uv} = g(h_u, h_v)$. Specifically, $g(\cdot)$ can be implemented as a simple inner product, cosine similarity, or a multi-layer perceptron (MLP), which outputs a probability

score indicating the likelihood of an edge existing between nodes $u$ and $v$ in the next snapshot. Then, the model is optimized via a Binary Cross-Entropy (BCE) loss, which penalizes the discrepancy between the predicted similarity scores and the ground-truth connectivity:

$$\mathcal{L} = - \sum_{(u,v) \in \mathcal{T}_t} \left( y_{uv} \log(\hat{y}_{uv}) + (1 - y_{uv}) \log(1 - \hat{y}_{uv}) \right),$$
(2)

where $\hat{y}_{uv} = s_{uv}$ and $\mathcal{T}_t$ denotes the set of edge labels in $\mathbf{A}_{t+1}^{train}$, *i.e.*, $y_{uv}$ is defined by the connectivity in the next snapshot. $y_{uv} = 1$ if a link exists between $u$ and $v$ at time $t+1$, and $y_{uv} = 0$ otherwise. Minimizing this loss forces the model $f_\theta(\cdot)$ to project the current structural information and historical temporal information into a space that is predictive of the future connectivity, allowing the model to capture the latent temporal-spatial evolution patterns. After that, the test graph $\mathbf{A}_t^{test}$ is input into the model to predict the subsequent snapshot. However, the effectiveness of the pretrained model is limited by natural structure shifts between $\mathbf{A}_t^{test}$ and $\mathbf{A}_t^{train}$, which disrupts the alignment between the backbone's learned patterns and the target statistics. This results in a distinct shift between the training and test-time distributions, as illustrated in Figure 4(a). Visualizations for additional datasets are available in Appendix B.3.2.

**Test-time Distribution Shift Prompts** To address the shifts, traditional TTA methods like Tent (Wang et al., 2021a) update batch normalization parameters via entropy

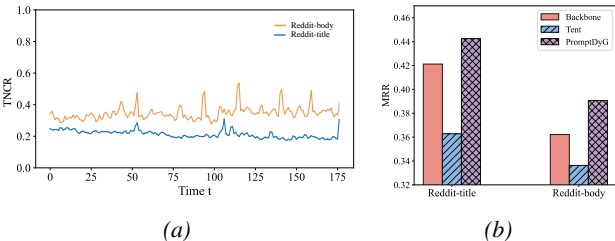

*(a)*          *(b)*

*Figure 5.* (a) Temporal Neighborhood Change Rate (TNCR) curves on two datasets, which quantifies the time dependence of neighborhood structures across consecutive snapshots (lower values indicate higher consistency and strong temporal dependence), more details can be found in the Appendix D. (b) Comparison of performance among backbone (Roland-GRU), Tent, and our PromptDyG.

minimization. However, in dynamic graph learning, this strategy would distort the pre-learned temporal dynamics essential for future prediction. As depicted in Figure 5, such methods can degrade the performance of the backbone, especially on datasets with strong temporal dependence, *e.g.*, Reddit-title and Reddit-body (Kumar et al., 2018).

Instead of utilizing model regularization as in existing TTA methods for static graphs, PromptDyG learns a dynamic prompt at each time step while keeping the DTDG backbone frozen. Specifically, at test-time step $t$, the graph prompt is designed as a set of learnable parameters, *i.e.*, $\mathbf{P}_t = [p_t^1, p_t^2, \cdots, p_t^N] \in \mathbb{R}^{N \times D}$. The adaptation embedding matrix $\widetilde{\mathbf{H}}_t$ is then obtained by injecting the prompt into the input feature space of the test graph and the pre-trained backbone $f_\theta^t$:

$$\widetilde{\mathbf{H}}_t = f_\theta^t(\mathbf{A}_t^{test}, \mathbf{X} + \mathbf{P}_t, \mathbf{H}_{t-1}). \tag{3}$$

By optimizing the prompt $\mathbf{P}_t$ with the DTDG backbone being frozen, PromptDyG achieves two key benefits. First, the model-agnostic, plug-and-play prompt effectively preserves the pre-learned temporal dynamics, ensuring they are not disturbed by short-time (current) structural noise. Second, the prompt essentially performs a non-linear shift in the latent space to align the structure-shifted test data with the model's pre-trained spatio-temporal evolution patterns (as illustrated in Lemma 4.2). As shown in Figure 4(b), Prompt-DyG successfully bridges this gap, where the training and test-time distributions span the same region.

In this paper, we instantiate the prompting framework by using Roland-GRU as the DTDG backbone. Notably, our framework's applicability extends beyond this backbone, as demonstrated by the experiments presented in Section 5.2, showcasing its versatility across various DTDG methods.

### 4.2. Prompt Adaptation via Unsupervised Feature-Wise Entropy Minimization

Link prediction typically assumes that nodes connected by edges should exhibit similar latent representations. To this

end, we operate on the latent embedding space and optimize the prompt $\mathbf{P}_t$ via unsupervised feature-wise entropy minimization at each inference phase. As shown in Figure 4(a), the structural shift causes the test-time distribution to deviate from the patterns learned during training, leading to indistinguishable similarity scores between positive and negative pairs (Figure 2(a)). This indicates that the backbone, suffering from structural shifts, tends to produce pattern-blurred and high-entropy representations, reflecting its uncertainty over the latent structures of the next time step. In contrast, a well-adapted model is expected to generate low-entropy, pattern-discriminative embeddings (Figure 2 (b)). To address this problem, we augment the feature of the test graph with the prompt $\mathbf{P}_t$ and optimize it by minimizing the unsupervised average Shannon entropy over the predicted distributions of the nodes, which is formulated as:

$$\min_{\mathbf{P}_t} \mathcal{L}_{ent} = -\frac{1}{|\mathcal{V}_s|} \sum_{i \in \mathcal{V}_s} \sigma(\tilde{h}_i) \log \sigma(\tilde{h}_i), \tag{4}$$

where $|\mathcal{V}_s|$ is the number of randomly sampled nodes and $\sigma(\cdot)$ is the softmax function. Note that this process requires no modifications to the pre-trained model, preserving the temporal dynamics encoded in the backbone and thus suitable for dynamic graph learning environments.

### 4.3. Theoretical Analysis

In this section, we provide a formal theoretical and empirical analysis to justify how the proposed feature-wise, label-free entropy minimization over the prompt matrix $\mathbf{P}_t$ enhances link prediction performance under structural shifts.

**Proposition 4.1.** *At each time step, the final linear layer can be written as $\widetilde{\mathbf{H}} = \hat{\mathbf{H}}\mathbf{W}$, where $\mathbf{W} = [\mu_1, \mu_2, \cdots, \mu_K] \in \mathbb{R}^{d \times K}$ is a learnable projection matrix in the backbone. Under the link prediction objective in Eq.(2), the resulting representation space lies in a $K$-dimensional subspace spanned by the columns of $\mathbf{W}$, where each column vector $\mu_k$ can be interpreted as a specific structural evolutionary prototype.*

The proof can be found in Appendix A. In the inference phase, the $k$-th component $\tilde{h}_{ik}$ of the node $i$'s embedding $\tilde{h}_i \in \mathbb{R}^{1 \times K}$ can be expressed as the dot product of $\hat{h}_i \in \mathbb{R}^{1 \times d}$ and the evolutionary prototype $\mu_k \in \mathbb{R}^{d \times 1}$. The probability of node $i$ belonging to pattern $k$ is given by the distribution $q_i = [q_{i1}, \ldots, q_{iK}]$, where $q_{ik} = \frac{\exp(\hat{h}_i\mu_k)}{\sum_{j=1}^{K} \exp(\hat{h}_i\mu_j)}$. Our objective is to optimize $\mathbf{P}_t$ at each time step $t$ by minimizing the Shannon entropy:

$$\min_{\mathbf{P}_t} \mathcal{L}_{ent} = -\frac{1}{|\mathcal{V}_s|} \sum_{i=1}^{|\mathcal{V}_s|} \sum_{k=1}^{K} q_{ik} \log q_{ik} \tag{5}$$

**Lemma 4.2.** *(Pattern Refinement). Assume that the input-output Jacobian $\mathbf{J}$ of the pre-trained backbone is non-*

*degenerate. Minimizing $\mathcal{L}_{ent}$ via gradient descent on $\mathbf{P}_t$ at each time step $t \in [1, T]$ forces the latent embedding $\hat{h}_i$ to converge toward its most relevant evolutionary prototype $\mu_{k^*}$. This ensures that the inference embedding $\tilde{h}_i$ achieves the maximal activation response toward the evolutionary prototype $\mu_{k^*}$, thus efficiently and continuously modeling the evolving patterns.*

The proof can be found in Appendix A. By minimizing the entropy with respect to the prompt $\mathbf{P}_t$, our method reshapes the prototype-assignment distribution $q_i$, which measures the alignment between node $i$ and the $K$ evolutionary prototype. The entropy gradient forces the distribution $q_i$ to approach a one-hot distribution (from high to low entropy), driving the embedding $\hat{h}_i$ to align tightly with the dominant prototype $\mu_{k^\star}$ while suppressing others (i.e., $\tilde{h}_{ik^*} \uparrow, \tilde{h}_{ij}(j \neq k^*) \downarrow$), thereby rectifying the shift-induced ambiguity and efficiently modeling the evolving patterns.

**Link Prediction Performance Enhancement.** We demonstrate that the pattern refinement directly leads to improved discriminative power for link prediction, as it relies on the similarity of node pairs, *i.e.*, $s_{uv} = g(\tilde{h}_u, \tilde{h}_v) = \tilde{h}_u \tilde{h}_v^\top$.

**Proposition 4.3.** *(Margin Expansion). Given a positive pair $(u, v) \in \mathcal{E}_{true}$ and a negative pair $(u, w) \notin \mathcal{E}_{true}$, optimizing $\mathbf{P}_t$ via entropy minimization increases the prediction margin $\gamma = s_{uv} - s_{uw}$.*

*Proof.* Due to temporal-spatial consistency (Wang et al., 2025), nodes $u$ and $v$ in a true link likely share the same evolutionary pattern $c$ (Zhu et al., 2016). According to Lemma 4.2, $\mathcal{L}_{ent} \rightarrow 0$ implies $\hat{h}_u \rightarrow \mu_c^\top$ and $\hat{h}_v \rightarrow \mu_c^\top$. Since the final embedding is obtained via linear projection $\tilde{h}_i = \hat{h}_i \mathbf{W}$, we have $\tilde{h}_u \approx \mu_c^\top \mathbf{W}$, $\tilde{h}_v \approx \mu_c^\top \mathbf{W}$, which implies $s_{uv} = \tilde{h}_u \tilde{h}_v^\top \approx \mu_c^\top \mathbf{W} \mathbf{W}^\top \mu_c$. According to Proposition 4.1, each column of $\mathbf{W}$ encodes a specific evolutionary prototype, and these prototypes are highly dissimilar and well-separated, yielding $\mathbf{W}^\top \mathbf{W} \approx \mathbf{I}_K$. Expressing the prototype vector as $\mu_c = \mathbf{W} e_c$ (where $e_c$ is the $c$-th standard basis vector), the score can be reformulated as $s_{uv} \approx (\mathbf{W} e_c)^\top \mathbf{W} \mathbf{W}^\top (\mathbf{W} e_c) = e_c^\top (\mathbf{W}^\top \mathbf{W})(\mathbf{W}^\top \mathbf{W}) e_c$. By substituting $\mathbf{W}^\top \mathbf{W} \approx \mathbf{I}_K$, we obtain $s_{uv} \approx e_c^\top (\mathbf{W}^\top \mathbf{W})^2 e_c \approx e_c^\top e_c = 1$, which maximizes the positive similarity.

Conversely, for a non-existent link $(u, w)$, the nodes typically belong to divergent patterns $c_1$ and $c_2$ ($c_1 \neq c_2$). Entropy minimization drives $\hat{h}_u \rightarrow \mu_{c_1}^\top$ and $\hat{h}_w \rightarrow \mu_{c_2}^\top$, yielding $\tilde{h}_u \approx \mu_{c_1}^\top \mathbf{W}$ and $\tilde{h}_w \approx \mu_{c_2}^\top \mathbf{W}$. The similarity score $s_{uw} = \tilde{h}_u \tilde{h}_w^\top \approx \mu_{c_1}^\top \mathbf{W} \mathbf{W}^\top \mu_{c_2}$. Similarly, since $\mu_{c_1} = \mathbf{W} e_{c_1}$ and $\mu_{c_2} = \mathbf{W} e_{c_2}$, the negative similarity score can be expanded as $s_{uw} \approx \mu_{c_1}^\top \mathbf{W} \mathbf{W}^\top \mu_{c_2} = e_{c_1}^\top (\mathbf{W}^\top \mathbf{W})(\mathbf{W}^\top \mathbf{W}) e_{c_2}$. By substituting $\mathbf{W}^\top \mathbf{W} \approx \mathbf{I}_K$, we obtain $s_{uw} \approx e_{c_1}^\top e_{c_2} = 0$ (since $c_1 \neq c_2$), which suppresses the similarity score.

Therefore, the prediction margin $\gamma = s_{uv} - s_{uw}$ increases after entropy minimization. $\square$

To support the theoretical analysis, we further conduct empirical studies on the UCI dataset to visualize the effectiveness of the prompt adaptation. As shown in Figure 2 (a), due to the distribution shift, the backbone produces highly overlapped similarity scores for positive and negative pairs, resulting in a small margin. In contrast, our adaptation increases positive-pair similarity while decreasing negative-pair similarity via pattern refinement (Figure 2 (b)). This enlarges the similarity margin and improves the model's generalization. Similar results on other datasets can be found in Appendix B.3.1.

### 4.4. Computational Complexity.

During inference, the computational complexity of entropy loss in PromptDyG is $O(SK)$ for each epoch. For prompt adaptation involving $e$ gradient steps (epochs) per snapshot, the cumulative complexity is $O(eSK)$, which is linear with respect to the number of sampled-nodes $S(S \ll N)$ and output embedding dimension $K$. Since the backbone is frozen, the memory overhead is negligible, involving only the lightweight prompt. In contrast, Tent (Wang et al., 2021a) optimizes a global entropy loss with a complexity of $O(NK)$, while the PIC loss in Matcha scales as $O(NK^2)$. Furthermore, both Tent and Matcha require higher memory costs to store intermediate activations for updating the pre-trained model parameters. As for GTrans, its contrastive loss exhibits a quadratic complexity of $O(N^2K)$. Additionally, its dependence on graph augmentation introduces computational latency and memory burdens. We empirically evaluate the efficiency of PromptDyG and those TTA methods in Subsection 5.4.

## 5. Experiments

### 5.1. Experimental Setup

**Datasets.** We conduct experiments on six datasets that have been extensively evaluated by previous studies for DTDGs, *i.e.*, AS-733 (Leskovec et al., 2005), Reddit-title (Kumar et al., 2018), Reddit-body (Kumar et al., 2018), UCI (Panzarasa et al., 2009), Bitcoin-OTC, and Bitcoin-Alpha (Kumar et al., 2016). Detailed statistics for those datasets are provided in Table 4 of Appendix B.1.

**Baselines.** We compare our PromptDyG to ten DTDG models, namely two versions of EvolveGCN (EvolveGCN-H and EvolveGCN-O), three versions of GCRN (GCRN-GRU, GCRN-LSTM), GCRN-Baseline, TGCN, three versions of Roland (Roland-Average, Roland-MLP, Roland-GRU), and the latest method SFDyG. We re-implement all DTDG baselines under the live-update evaluation setting

*Table 1.* Comparison of MRR (adaptation time) results on six datasets. We report the mean of MRR over three runs, with the best results highlighted in bold and green, and the second-best results highlighted in purple. Avg. Rank and Avg. MRR show the average MRR and ranking of each method across all datasets, respectively. The values in parentheses represent the average adaptation time per epoch (ms).

| Methods | Datasets | | | | | | Avg. Rank | Avg. MRR |
|---|---|---|---|---|---|---|---|---|
| | AS-733 | Reddit-title | Reddit-body | UCI | Bitcoin-OTC | Bitcoin-Alpha | | |
| | DTDG models | | | | | | | |
| EvolveGCN-H | 0.3004 | 0.1233 | 0.0745 | 0.0886 | 0.0957 | 0.0805 | 12.17 | 0.1272 |
| EvolveGCN-O | 0.1799 | 0.1499 | 0.0685 | 0.0639 | 0.0129 | 0.0249 | 13.67 | 0.0833 |
| GCRN-GRU | 0.3443 | 0.3420 | 0.2190 | 0.0999 | 0.1726 | 0.1453 | 8.33 | 0.2205 |
| GCRN-LSTM | 0.3411 | 0.3499 | 0.2194 | 0.0946 | 0.1744 | 0.1441 | 8.50 | 0.2206 |
| GCRN-Baseline | 0.3363 | 0.3556 | 0.2193 | 0.0865 | 0.1789 | 0.1456 | 9.00 | 0.2204 |
| TGCN | 0.3440 | 0.3915 | 0.2503 | 0.0627 | 0.0845 | 0.0749 | 9.83 | 0.2013 |
| SFDyG | 0.3391 | 0.4219 | 0.3271 | 0.1143 | 0.1804 | 0.1583 | 4.83 | 0.2569 |
| Roland-Average | 0.2749 | 0.3565 | 0.2868 | 0.0694 | 0.1215 | 0.0925 | 10.50 | 0.2003 |
| Roland-MLP | 0.3008 | 0.3947 | 0.2689 | 0.0996 | 0.1612 | 0.1313 | 8.83 | 0.2261 |
| Roland-GRU | 0.3381 | 0.4212 | 0.3622 | 0.1065 | 0.1932 | 0.1547 | 4.83 | 0.2627 |
| Roland-GRU | TTA-based models | | | | | | | |
| +Tent | 0.3135(6.71) | 0.3628(39.72) | 0.3363(8.55) | 0.1225(19.20) | 0.1941(8.95) | 0.1637(5.45) | 5.50 | 0.2488 |
| +Matcha | 0.3206(17.57) | 0.4091(63.89) | 0.3445(27.39) | 0.1041(45.68) | 0.1955(22.56) | 0.1675(14.07) | 4.67 | 0.2569 |
| +GTrans | 0.3394(26.25) | 0.4276(149.83) | 0.3712(60.26) | 0.1064(29.80) | 0.1942(24.87) | 0.1651(15.46) | 3.33 | 0.2673 |
| PromptDyG | **0.3448**(3.85) | **0.4425**(33.90) | **0.3906**(6.81) | **0.1248**(9.15) | **0.2004**(4.97) | **0.1732**(3.20) | 1.00 | 0.2794 |

*Table 2.* Results of enabling existing DTDG methods with PromptDyG. Bold denotes better performance. Values in parentheses represent the improvement over the backbone.

| Method | Datasets | | | | | |
|---|---|---|---|---|---|---|
| | AS-733 | Reddit-title | Reddit-body | UCI | Bitcoin-OTC | Bitcoin-Alpha |
| EvolveGCN-H | 0.3004 | 0.1233 | 0.0745 | 0.0886 | 0.0957 | 0.0805 |
| +PromptDyG | **0.3189**(+1.85%) | **0.1570**(+3.37%) | **0.1067**(+3.22%) | **0.1018**(+1.32%) | **0.1189**(+2.32%) | **0.0980**(+1.75%) |
| GCRN-GRU | 0.3443 | 0.3420 | 0.2190 | 0.0999 | 0.1726 | 0.1453 |
| +PromptDyG | **0.3454**(+0.11%) | **0.3589**(+1.69%) | **0.2331**(+1.41%) | **0.1367**(+3.68%) | **0.1912**(+1.86%) | **0.1562**(+1.09%) |
| GCRN-Baseline | 0.3363 | 0.3556 | 0.2193 | 0.0865 | 0.1789 | 0.1456 |
| +PromptDyG | **0.3397**(+0.34%) | **0.3704**(+1.48%) | **0.2326**(+1.33%) | **0.1181**(+3.16%) | **0.1872**(+0.83%) | **0.1523**(+0.67%) |
| TGCN | 0.3440 | 0.3915 | 0.2503 | 0.0627 | 0.0845 | 0.0749 |
| +PromptDyG | **0.3455**(+0.15%) | **0.4032**(+1.17%) | **0.2640**(+1.37%) | **0.0779**(+1.52%) | **0.1025**(+1.80%) | **0.0922**(+1.73%) |
| SFDyG | 0.3391 | 0.4219 | 0.3271 | 0.1143 | 0.1804 | 0.1583 |
| +PromptDyG | **0.3425**(+0.34%) | **0.4322**(+1.03%) | **0.3390**(+1.19%) | **0.1270**(+1.27%) | **0.1901**(+0.97%) | **0.1664**(+0.81%) |

using the official public code[1]. Additionally, we compare PromptDyG with representative TTA approaches, including a prominent general TTA method, Tent, as well as two graph-specific TTA methods: GTrans (graph adaptation) and Matcha (model adaptation). A comprehensive description of these baselines can be found in Appendix B.2.

**Evaluation Metric.** Following Roland, we evaluate our model using the Mean Reciprocal Rank (MRR). Specifically, for each node $u$ that forms a positive edge $(u, v)$ at time $t + 1$, we randomly sample 1000 negative edges originating from $u$ and compute the rank of the prediction score of $(u, v)$ among these candidates. The final MRR is obtained by averaging the reciprocal ranks over all such nodes $u$. Additionally, we also report two other commonly used metrics: the Area Under the Receiver Operating Characteristic Curve (AUROC) and F1-Score in Appendix B.3.3.

[1] https://github.com/snap-stanford/roland

**Implementation Details.** We follow the live-update evaluation protocol proposed, where model performance is continuously evaluated over all available temporal snapshots. For snapshot at each time $t$, edges are randomly split into training $\mathbf{A}_t^{train}$, validation $\mathbf{A}_t^{val}$, and test sets $\mathbf{A}_t^{test}$ with an 8:1:1 ratio. The hyperparameters of baselines are set according to those provided in Roland. Note that PromptDyG only introduces two hyperparameters, including the learning rate and the number of epochs. We optimize the prompt to minimize the Shannon entropy over 1, 5, 10, or 20 steps, using the Adam optimizer with the learning rate selected from $\{0.01, 0.001, 0.005, 0.0005\}$ based on the unsupervised entropy loss of Eq.(4). All experiments are executed on a GeForce RTX 3090 GPU (24 GB).

### 5.2. Main Results

**Comparing to SOTA DTDG Methods.** The temporal link prediction results against different DTDG models are shown

in Table 1. Firstly, we observe that our method achieves the best performance on all six dynamic benchmarks, leading to the top average rank and the highest average MRR across datasets. Secondly, compared with the backbone Roland-GRU, our method yields consistent improvements on all six datasets, with the largest improvement on Reddit-body, which obtains a gain of 2.84%. These results indicate that explicitly adapting to structural shifts can enhance the model's generalization capabilities, enabling it to maintain high predictive accuracy of subsequent snapshot prediction.

**Comparing to TTA Methods.** We further evaluate our PromptDyG with representative TTA methods. From Table 1, we find that Tent and Matcha underperform the backbone model (Roland-GRU) on AS-733, Reddit-body, and Reddit-title. This is primarily because these datasets exhibit strong temporal dependence, as evidenced in Appendix D. The backbone effectively captures historical evolution patterns by jointly leveraging GNNs and RNNs. However, Tent and Matcha update well-trained model parameters online, which inadvertently disturbs the pretrained temporal representations and breaks the encoded historical dependencies, leading to degraded performance in future predictions. In contrast, the graph-adaptive TTA method GTrans and our PromptDyG freeze the pretrained backbone and introduce data-centric learnable parameters to cope with structural shifts, achieving improved performance on these three datasets. For the remaining datasets, where no clear or stable evolution patterns can be observed (see Figure 11 in Appendix D), learning meaningful temporal representations becomes considerably more challenging. In such scenarios, introducing inference-time adaptation provides an effective form of on-the-fly regularization, helping the model adjust to rapidly changing structures. Consequently, all TTA methods consistently yield favorable performance. However, the average adaptation times in parentheses show that these TTA baselines suffer from substantial latency during the adaptation phase, especially for the augmentation-based method GTrans, where graph data augmentation and contrastive learning objectives bring heavy computational overhead (additional evidence is illustrated in Figure 8).

Overall, our PromptDyG framework is more efficient for tackling structural shifts during inference in dynamic graphs. It enlarges the similarity margin between positive and negative sample pairs through optimizing lightweight learnable prompts and consistently improves temporal link prediction performance across different datasets.

**Enabling Existing DTGD Methods with PromptDyG.** PromptDyG can be devised as a plug-and-play module to tackle test-time structural shifts of existing DTDG models. Table 2 reports the results of equipping multiple representative DTDG backbones with our PromptDyG. We observe consistent improvements across all backbones and datasets.

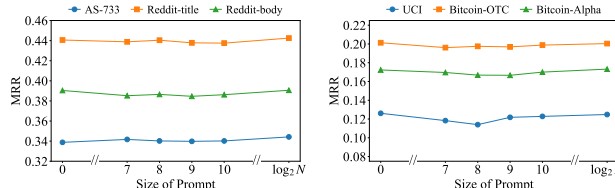

*Figure 6.* The MRR results w.r.t. the size of the graph prompts. The $x$-axis denotes the exponent $2^i$.

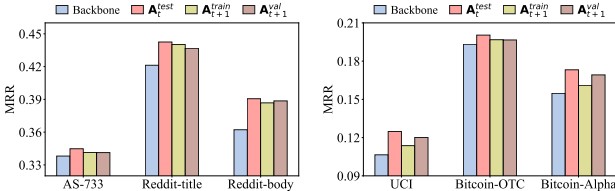

*Figure 7.* The MRR results of PromptDyG w.r.t. different inputs for prompt adaptation.

For example, when applied to GCRN-GRU, our method yields the largest absolute improvement of 3.68% on UCI. Similarly, on the relatively weaker backbone EvolveGCN-H, PromptDyG substantially boosts performance on Reddit-title by 3.37% and on Reddit-body by 3.22%. Even for stronger backbones such as SFDyG, PromptDyG still delivers stable improvements and achieves the best results on Reddit-title/body and Bitcoin-Alpha. In summary, while existing DTDGs models excel at capturing spatio-temporal dynamics during training, they often overlook test-time structural shifts. By explicitly mitigating the shift, PromptDyG refines the misaligned representations and consistently enhances temporal link prediction performance.

### 5.3. Ablation Study

**Sensitivity w.r.t. the Size of Prompts.** We evaluate the sensitivity of PromptDyG w.r.t the size of the graph prompt. We vary the prompt size from a shared global vector (*i.e.*, $p \in \mathbb{R}^{1 \times D}$) to node-specific parameters (*i.e.*, $\mathbf{P} \in \mathbb{R}^{N \times D}$). For intermediate sizes ($m \in \{128, \cdots, 1024\}$), we adopt a low-rank factorization strategy following GPF (Fang et al., 2024), where the prompt dimension is controlled via a projection matrix $\mathbf{W} \in \mathbb{R}^{D \times m}$. Therefore, the size of the prompt $\mathbf{P}$ are set to $s \times D$, where $s \in \{1, 128, 256, 512, 1024, N\}$.

Figure 6 illustrates the MRR performance across different prompt sizes, where we group the datasets into two figures based on the magnitude of the MRR values to better highlight the performance gaps. As shown in the results, the performance curves are essentially flat and exhibit remarkable robustness of PromptDyG to prompt size. Therefore, for large-scale graph datasets, employing a compact global prompt is a viable strategy to maximize scalability without compromising predictive performance.

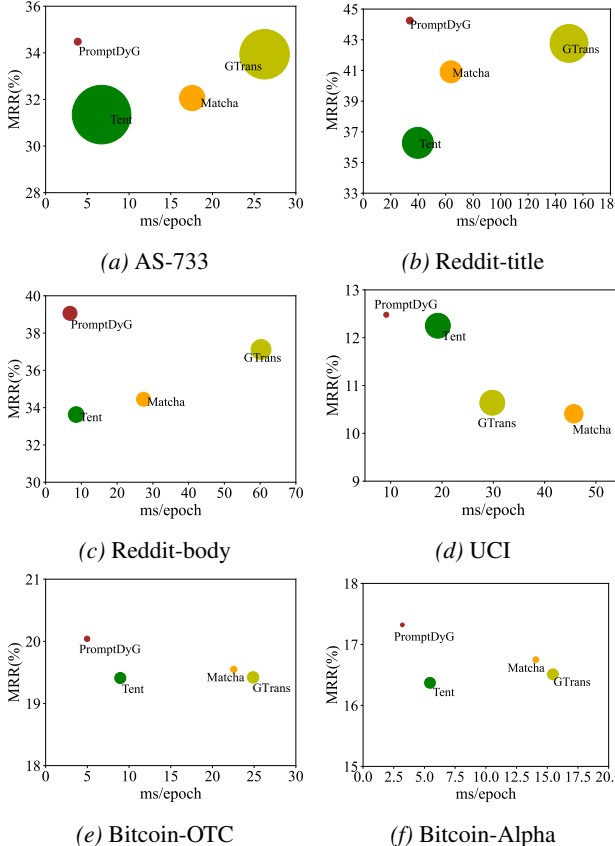

*Figure 8.* Comparison in terms of MRR(%), average adaptation time (per epoch), and relative maximum GPU memory consumption. The x-axis denotes the average adaptation time, the y-axis denotes MRR, and the size of the circle indicates the GPU memory consumption.

**Sensitivity w.r.t. Different Inputs for Adaptation.** We further analyze the sensitivity of our framework w.r.t. the source of the structural input used for prompt optimization. Since the future topologies of the training and validation splits ($\mathbf{A}_{t+1}^{train}$ and $\mathbf{A}_{t+1}^{val}$) are technically accessible as ground-truth labels during the pre-training phase, one might consider using them for adaptation.

As shown in Figure 7, we observe that while exploiting the accessible future structures improves performance, utilizing the test structure consistently yields the best performance, confirming the necessity of directly adapting to the specific structural distribution of the test data to mitigate inference-time shifts.

### 5.4. Complexity Comparison

In this subsection, we comprehensively compare Prompt-DyG to other TTA models concerning MRR, the average adaptation time per epoch, and GPU memory consumption during the test-time on six datasets.

As shown in Figure 8, our PromptDyG achieves the best performance with low adaptation time and memory overhead. GTrans incurs significant computational overhead and memory consumption, which can be attributed to its expensive graph data augmentation and contrastive learning objectives. Similarly, Tent and Matcha exhibit high resource demands as they necessitate backpropagation to update the backbone parameters during the adaptation process. In contrast, our method freezes the backbone and only updates the prompts, consistently achieving the lowest average adaptation time across all datasets and minimizing GPU memory overhead.

### 6. Conclusion

In this paper, we propose a novel framework, PromptDyG, for discrete-time dynamic graph learning. Specifically, we adapt the live-update evaluation setting as graph snapshots emerge sequentially, and the models are evaluated in a continual manner. Existing works typically utilize static GNN models to represent each graph snapshot, and a temporal module is designed to capture temporal dynamics for subsequent snapshot prediction. However, these methods overlook the structural shifts between training and test snapshots, leading to inferior performance.

To address this problem, we propose to enhance the generality of the learned model at each time step via unsupervised test-time prompt adaptation. To preserve the temporal dependencies, a lightweight learnable graph prompting mechanism is introduced while keeping the backbone frozen. Theoretical and empirical results demonstrate that the proposed PromptDyG effectively enlarges the similarity margin between positive and negative node pairs under structural shifts for subsequent graph snapshot predictions. Moreover, the proposed method could act as a plug-and-play module, consistently improving different dynamic graph backbones.

A limitation of this work is that it focuses solely on the link prediction task in DTDGs. Future work will explore other data types, such as CTDGs, and extend our framework to various tasks, such as dynamic graph anomaly detection.

### Acknowledgments

This research is supported in part by the Singapore Ministry of Education (MOE) Academic Research Fund (AcRF) Tier 1 Grant (24-SIS-SMU-008) and the Lee Kong Chian Fellowship (T050273).

### Impact Statement

This paper presents work whose goal is to advance the field of graph machine learning. There are many potential societal consequences of our work, none of which we feel must be specifically highlighted here.

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

# A. Theoretical Analysis

**Proposition A.1.** *At each time step, the final linear layer can be written as $\widetilde{\mathbf{H}} = \hat{\mathbf{H}}\mathbf{W}$, where $\mathbf{W} = [\mu_1, \mu_2, \cdots, \mu_K] \in \mathbb{R}^{d \times K}$ is a learnable projection matrix in the backbone. Under the link prediction objective in Eq. (2), the resulting representation space lies in a $K$-dimensional subspace spanned by the columns of $\mathbf{W}$, where each column vector $\mu_k$ can be interpreted as a specific structural evolutionary prototype.*

*Proof.* Since $\widetilde{\mathbf{H}} = \hat{\mathbf{H}}\mathbf{W}$, each node embedding satisfies $\tilde{h}_i = \hat{h}_i \mathbf{W}$, which implies that every embedding is a linear combination of the column vectors of $\mathbf{W}$. Given that $\mathbf{W} = [\mu_1, \mu_2, \cdots, \mu_K]$, then $\tilde{h}_i = \sum_{k=1}^{K} \alpha_{ik} \mu_k$ for some coefficients $\alpha_{ik}$, indicating that all embeddings lie in the subspace $\text{span}(\mathbf{W}) = \text{span}(\mu_1, \mu_2, \cdots, \mu_K)$. Given the link probability between nodes $i$ and $j$ defined as $s_{ij} = (\tilde{h}_i \tilde{h}_j^\top)$, substituting $\tilde{h}_i = \hat{h}_i \mathbf{W}$ yields $s_{ij} = (\hat{h}_i \mathbf{W}\mathbf{W}^\top \hat{h}_j^\top)$ or in matrix form $\mathbf{S} = (\hat{\mathbf{H}}\mathbf{W}\mathbf{W}^\top \hat{\mathbf{H}}^\top)$. Minimizing the link prediction loss is therefore equivalent to fitting $\mathbf{A} \approx (\hat{\mathbf{H}}\mathbf{W}\mathbf{W}^\top \hat{\mathbf{H}}^\top)$, which can be interpreted as reconstructing the adjacency matrix. Furthermore, let $\mathbf{M} = \mathbf{W}\mathbf{W}^\top$, since $\mathbf{W} \in \mathbb{R}^{d \times K}$, we have $\text{rank}(\mathbf{M}) \le K$, leading to a low-rank interaction model $\mathbf{A} \approx (\hat{\mathbf{H}}\mathbf{M}\hat{\mathbf{H}}^\top)$. Such low-rank graph models capture a finite number of dominant interaction patterns (Hoff et al., 2002). Accordingly, the embedding space is confined to the subspace spanned by $\{\mu_1, \mu_2, \cdots, \mu_K\}$, where each column vector $\mu_k$ represents one prototypical direction describing a characteristic interaction pattern, and so the node embedding $\tilde{h}_i = \hat{h}_i \mathbf{W}$ can be interpreted as the projection of node $i$ onto these prototype directions. Therefore, the latent representation space admits a set of structural evolutionary prototypes corresponding to the columns of $\mathbf{W}$.

Secondly, we provide an empirical justification for this proposition. We calculate pairwise cosine similarities of the $K$ prototypes on six datasets. As shown in Table 3, these exceptionally low mean inter-prototype cosine similarities (excluding the diagonal entries) indicate that the prototypes are highly dissimilar and well-separated, effectively precluding mode collapse and capturing distinct evolutionary patterns.

$\square$

*Table 3.* The mean±std of inter-prototype cosine similarity on six datasets.

| AS-733 | Reddit-title | Reddit-body | UCI | Bitcoin-OTC | Bitcoin-Alpha |
|--------|--------------|-------------|-----|-------------|---------------|
| 0.0013±0.0851 | 0.0008±0.0903 | 0.0007±0.0905 | 0.0003±0.0810 | 0.0030±0.0800 | 0.0014±0.0883 |

**Lemma A.2.** *(Pattern Refinement). Assume that the input-output Jacobian $\mathbf{J}$ of the pre-trained backbone is non-degenerate. Minimizing $\mathcal{L}_{ent}$ via gradient descent on $\mathbf{P}_t$ at each time step $t \in [1, T]$ forces the latent embedding $\hat{h}_i$ to converge toward its most relevant evolutionary prototype $\mu_{k^*}$. This ensures that the inference embedding $\tilde{h}_i$ achieves the maximal activation response toward the evolutionary prototype $\mu_{k^*}$, thus efficiently and continuously modeling the evolving patterns.*

*Proof.* For the test snapshot at time $t$, the frozen DTDG backbone produces node embeddings

$$\widetilde{\mathbf{H}}_t = f_\theta^t(\mathbf{A}_t^{test}, \mathbf{X} + \mathbf{P}_t, \mathbf{H}_{t-1}), \tag{6}$$

where $\mathbf{P}_t$ is the only learnable variable at inference time. $K$ evolutionary prototypes $\{\mu_k\}_{k=1}^{K}$ (implicitly encoded by the pre-trained latent space) are introduced by the final linear layer. For each node $i$, the prototype logits and probability distribution are

$$\tilde{h}_{ik} = \hat{h}_i \mu_k, \qquad q_{ik} = \text{Softmax}(\tilde{h}_i)_k. \tag{7}$$

The per-node entropy objective is

$$\mathcal{L}_i = -\sum_{k=1}^{K} q_{ik} \log q_{ik} = H(q_i), \tag{8}$$

and the full loss is $\mathcal{L}_{ent} = \frac{1}{|\mathcal{V}_s|} \sum_{i \in \mathcal{V}_s} \mathcal{L}_i$, where $\mathcal{V}_s$ is sampled node set.

**Step 1: Gradient w.r.t. logits yields a sharpening direction.** Let $H(q_i)$ denote the entropy of $q_i$ at the current iterate. Using the Softmax Jacobian $\frac{\partial q_{ij}}{\partial \tilde{h}_{ik}} = q_{ij}(\mathbb{I}[j = k] - q_{ik})$, we obtain

$$
\begin{aligned}
\frac{\partial \mathcal{L}_i}{\partial \tilde{h}_{ik}} &= -\sum_{j=1}^{K} \frac{\partial q_{ij}}{\partial \tilde{h}_{ik}} \big( \log q_{ij} + 1 \big) \\
&= -\sum_{j=1}^{K} q_{ij}(\mathbb{I}[j = k] - q_{ik}) \big( \log q_{ij} + 1 \big) \\
&= -q_{ik} \Big( \log q_{ik} - \sum_{j=1}^{K} q_{ij} \log q_{ij} \Big) \\
&= -q_{ik} \big( \log q_{ik} + H(q_i) \big).
\end{aligned}
\tag{9}
$$

Hence, a gradient descent step on $\tilde{h}_{ik}$ would be

$$
\tilde{h}_{ik}^{(s+1)} = \tilde{h}_{ik}^{(s)} - \eta \frac{\partial \mathcal{L}_i}{\partial \tilde{h}_{ik}} = \tilde{h}_{ik}^{(s)} + \eta \, q_{ik}^{(s)} \big( \log q_{ik}^{(s)} + H(q_i^{(s)}) \big).
\tag{10}
$$

Define the entropy-dependent threshold $\tau_i \triangleq \exp(-H(q_i^{(s)}))$. Specifically, $\log q_{ik}^{(s)} + H(p_i^{(s)}) > 0$ holds if and only if $q_{ik}^{(s)} > \tau_i$. Conversely, $\log q_{ik}^{(s)} + H(q_i^{(s)}) < 0$ holds if and only if $q_{ik}^{(s)} < \tau_i$. This implies

$$
q_{ik}^{(s)} > \tau_i \Rightarrow \tilde{h}_{ik}^{(s+1)} > \tilde{h}_{ik}^{(s)}, \qquad q_{ik}^{(s)} < \tau_i \Rightarrow \tilde{h}_{ik}^{(s+1)} < \tilde{h}_{ik}^{(s)}.
\tag{11}
$$

Therefore, entropy minimization increases the logits of higher-probability prototypes ($\mu_{k^*}$) and suppresses those of lower-probability ones, making $q_i$ more peaked and asymptotically approaching a one-hot distribution from high to low entropy (i.e., $\tilde{h}_{ik^*} \uparrow, \tilde{h}_{ij}(j \neq k^*) \downarrow$).

**Step 2: Transferring the sharpening direction to embeddings.** By the chain rule with $\tilde{h}_{ik} = \hat{h}_i \mu_k$, we have

$$
\frac{\partial \mathcal{L}_i}{\partial \tilde{h}_i} = \sum_{k=1}^{K} \frac{\partial \mathcal{L}_i}{\partial \tilde{h}_{ik}} \frac{\partial \tilde{h}_{ik}}{\partial \hat{h}_i} = \sum_{k=1}^{K} \frac{\partial \mathcal{L}_i}{\partial \tilde{h}_{ik}} \mu_k = -\sum_{k=1}^{K} q_{ik} \big( \log q_{ik} + H(q_i) \big) \mu_k.
\tag{12}
$$

Eq. (11) shows that the prototype(s) with relatively larger probability receive a positive drive through $\log q_{ik} + H(q_i) > 0$, whereas low-probability prototypes receive a negative drive. Thus, the embedding update induced by descending $\mathcal{L}_i$ is biased toward increasing alignment with the dominant prototype(s), which corresponds to moving $\hat{h}_i$ toward its most relevant evolution pattern $\mu_{k^*}$.

**Step 3: Converting the embedding-level direction into a prompt update.** Since $\mathbf{P}_t$ affects $\mathcal{L}_{ent}$ only through the frozen mapping $\tilde{h}_i$, the gradient w.r.t. $\mathbf{P}_t$ is given by

$$
\nabla_{\mathbf{P}_t} \mathcal{L}_{ent}(\mathbf{P}_t) = \frac{1}{|\mathcal{V}_s|} \sum_{i \in \mathcal{V}_s} \big( \mathbf{J}_i(\mathbf{P}_t) \big)^\top \frac{\partial \mathcal{L}_i}{\partial \tilde{h}_i}, \qquad \mathbf{J}_i(\mathbf{P}_t) \triangleq \frac{\partial \tilde{h}_i(\mathbf{P}_t)}{\partial \mathbf{P}_t},
\tag{13}
$$

Particularly, the input-output Jacobian of the backbone is defined as $\mathbf{J} = \frac{\partial \widetilde{\mathbf{H}}_t}{\partial \mathbf{P}_t}$, where $\widetilde{\mathbf{H}}_t = f_\theta(\mathbf{P}_t) \in \mathbb{R}^{N \times K}$ is the final output. The inference-time adaptation step is $\mathbf{P}_t^{(s+1)} = \mathbf{P}_t^{(s)} - \eta \nabla_{\mathbf{P}_t} \mathcal{L}_{ent}$. Given that the Jacobian $\mathbf{J}$ is non-degenerate, the sharpening direction in Eq. (12) is propagated to $\mathbf{P}_t$, thereby updating the input features $\mathbf{X} + \mathbf{P}_t$ to increase the dominant logit(s) and suppress the others (Eq. (11)). Consequently, minimizing $\mathcal{L}_{ent}$ via gradient descent on $\mathbf{P}_t$ drives $\tilde{h}_i$ toward its most relevant evolutionary prototype $\mu_{k^*}$ in the pre-trained latent space.

Furthermore, we provide the proof of the assumption that the input-output Jacobian $\mathbf{J}$ of the pre-trained backbone is non-degenerate. To facilitate a rigorous analysis within the framework of linear operators, we employ the vectorization operator $\text{vec}(\cdot)$ to recast $\mathbf{J}$ into a standard matrix form: $\mathbf{J} = \frac{\partial \text{vec}(\widetilde{\mathbf{H}}_t)}{\partial \text{vec}(\mathbf{P}_t)} \in \mathbb{R}^{NK \times ND}$. Then, since the prompt parameters provide $ND$

degrees of freedom for $NK$-dimensional output, $\mathbf{J}$ is a wide matrix ($ND > NK$). Leveraging over-parameterization theory and Neural Tangent Kernel (NTK) theory (Jacot et al., 2018; Du et al., 2018), such over-parameterized regimes ensure that the Gram matrix $\mathbf{JJ}^\top$ remains strictly positive-definite during optimization. Consequently, the full-row-rank property of $\mathbf{J}$ is structurally guaranteed, ensuring that gradient signals are effectively back-propagated to the prompt space.

Additionally, this assumption can be justified by our empirical evidence. A properly pre-trained backbone can learn semantically meaningful node representations from input data; however, a low-rank Jacobian would imply feature collapse, which contradicts the model's observed efficacy. Consequently, a well-functioning pre-trained model provides a guarantee that its input-output Jacobian is neither ill-conditioned nor rank-deficient (non-degenerate). This is verified by t-SNE visualizations (Figs. 4 and 10), where the successful prompt-based realignment of latent representations confirms an effective gradient flow.

$\square$

## B. Experiment Details

*Table 4.* Summary of Dataset statistics.

| Datasets | #Edges | #Nodes | Range | Snapshot Frequency | #Snapshots |
|---|---|---|---|---|---|
| AS-733 | 11,965,533 | 7,716 | Nov 8, 1997 - Jan 2, 2000 | daily | 733 |
| Reddit-title | 571,927 | 54,075 | Dec 31, 2013 - Apr 30, 2017 | weekly | 178 |
| Reddit-body | 286,561 | 35,776 | Dec 31, 2013 - Apr 30, 2017 | weekly | 178 |
| UCI | 59,835 | 1,899 | Apr 15, 2004 - Oct 26, 2004 | weekly | 29 |
| Bitcoin-OTC | 35,592 | 5,881 | Nov 8, 2010 - Jan 24, 2016 | weekly | 279 |
| Bitcoin-Alpha | 24,186 | 3,783 | Nov 7, 2010 - Jan 21, 2016 | weekly | 274 |

### B.1. Description of Datasets

- **Autonomous systems.** The autonomous systems (AS-733) dataset[2] is a router-level communication network dataset, where routers exchange traffic flows with peers, and it is commonly used for forecasting future message exchanges.

- **Reddit-title and Reddit-body.** Reddit-title and Reddit-body[3] are directed subreddit-to-subreddit hyperlink networks constructed from hyperlinks appearing in post titles and bodies, respectively.

- **UC Irvine messages.** UCI-Messages (UCI)[4] is an online student social network from the University of California, Irvine, where links represent private messages exchanged between users.

- **Bitcoin-OTC and Bitcoin-Alpha.** Bitcoin-OTC[5] and Bitcoin-Alpha[6] are who-trusts-whom networks of Bitcoin traders from the OTC and Alpha platforms, respectively, where users rate each other based on trust, and the datasets are commonly used for rating polarity prediction and future interaction forecasting.

### B.2. Description of Baselines

- **EvolveGCN** (Pareja et al., 2020) captures temporal dynamics by evolving the GCN parameters (weight matrices) via a Recurrent Neural Network (RNN), rather than learning temporal node embeddings. This design allows the model to handle dynamic graphs with frequent changes in node sets (inductive setting). The method includes two architectures: EvolveGCN-H, which extends a GRU by treating GCN weights as the hidden states, and EvolveGCN-O, which utilizes an LSTM to model the evolution of weight matrices directly.

- **TGCN** (Zhao et al., 2019) is a traffic forecasting model that integrates GCN and gated recurrent units to capture spatial and temporal dependencies jointly.

---

[2] http://snap.stanford.edu/data/as-733.html
[3] http://snap.stanford.edu/data/soc-RedditHyperlinks.html
[4] http://konect.uni-koblenz.de/networks/opsahl-ucsocial
[5] http://snap.stanford.edu/data/soc-sign-bitcoin-otc.html
[6] http://snap.stanford.edu/data/soc-sign-bitcoin-alpha.html

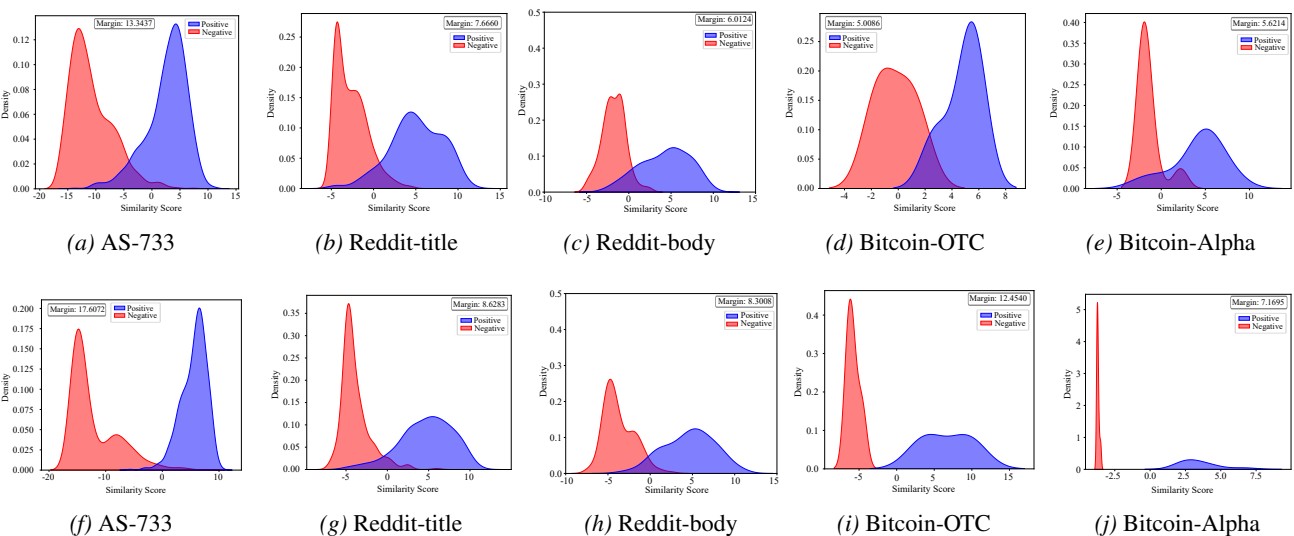

*Figure 9.* Similarity score distribution for positive and negative node pairs from backbone (the first row) and PromptDyG (the second row) on five datasets. For each dataset, the similarity is computed at the same test snapshot for both methods.

- **GCRN** (Seo et al., 2018) uses ChebNet (Defferrard et al., 2016) for spatial modeling and employs GRU (GCRN-GRU) or LSTM (GCRN-GRU) to capture temporal dependencies, with separate GNN modules employed to compute the different gates of the recurrent unit. GCRN-Baseline first applies spectral graph convolution for spatial feature extraction and then uses a standard LSTM for temporal modeling.

- **Roland** (You et al., 2022) proposes a live-update evaluation protocol and transforms static GNNs into dynamic ones by recurrently updating hierarchical node states over time. Based on different designs of the update module, ROLAND proposes three variants: Roland-GRU, Roland-LSTM, and Roland-Average.

- **SFDyG** (Qi et al., 2025) is a recently proposed method that focuses on modeling temporal edges in discrete-time dynamic graphs. By fusing multiple input snapshots into a unified temporal graph and incorporating Hawkes processes, SFDyG captures temporal and structural patterns efficiently while reducing the computational overhead associated with sequential modeling.

- **Tent** (Wang et al., 2021a) proposes a fully test-time adaptation framework, which updates model parameters of batch normalization by minimizing prediction entropy.

- **GTrans** (Jin et al., 2023) performs test-time adaptation by augmenting the target graph and optimizing a contrastive objective, where positive views are generated via drop edges or nodes, and negative samples are constructed by perturbing node features.

- **Matcha** (Bao et al., 2025) adapts pre-trained GNNs by adjusting hop-aggregation parameters to accommodate changes in node connectivity and introduces a prediction-informed clustering loss to enhance representation quality. Moreover, Matcha can be combined with existing TTA methods to jointly handle structure and attribute shifts, achieving robust performance under diverse distribution shift settings. In this work, we implement Matcha by adding an aggregation layer with learnable parameters to the backbone GNN. Moreover, to accommodate its class-aware loss, we accordingly reduce the dimension of the latent representations.

## B.3. Additional Experimental Results

### B.3.1. SIMILARITY SCORE DISTRIBUTION ANALYSIS ON ADDITIONAL DATASETS

In this subsection, we extend the similarity score and margin analysis to the remaining datasets. Following the same protocol as in Figure 2, we compute similarity scores for positive and negative node pairs from (i) direct inference with the frozen backbone and (ii) our PromptDyG at the same test snapshot. We then evaluate positive/negative pair similarity distributions

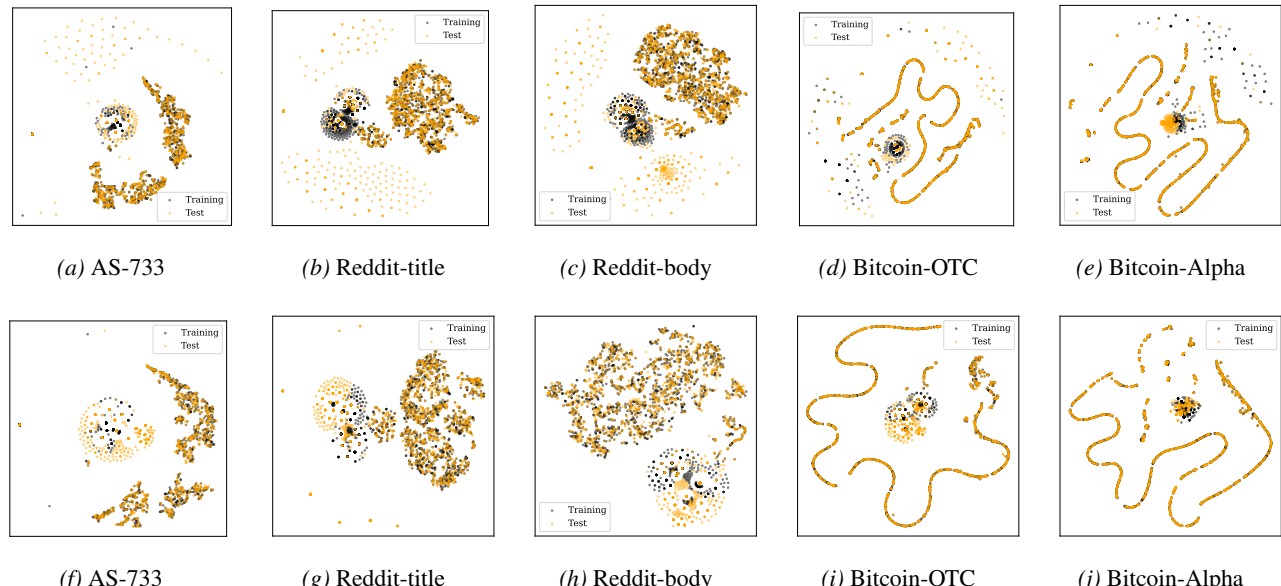

*Figure 10.* t-SNE visualization of training and test-time embeddings from the backbone (top row) and PromptDyG (bottom row) across five datasets. For each dataset, the embeddings are visualized at a specific time step.

and report the margin. As shown in Figure 9, across all additional datasets, our method consistently increases positive-pair similarities while suppressing negative-pair similarities, yielding a larger margin and corroborating the effectiveness of PromptDyG under inference-time structural shifts.

### B.3.2. T-SNE VISUALIZATION ON ADDITIONAL DATASETS

In this subsection, we extend the t-SNE visualization to the remaining datasets. To ensure visual clarity given the large number of nodes, we randomly sampled 2,000 nodes for visualization on those datasets. As shown in Figure 10, we observe that the structural distribution shift prevents the test-time representations from aligning with the patterns learned by the backbone (Roland-GRU) during training, resulting in a separation between the two distributions and evidencing the model's failure to generalize. In contrast, PromptDyG successfully bridges this distributional gap, demonstrating that the training and test-time embeddings occupy the same latent region, which indicates effective adaptation to the structural shifts.

### B.3.3. ADDITIONAL PERFORMANCE METRICS

In the main text (Table 1 and Table 2), we primarily reported the MRR to evaluate the performance of the predicted links. To provide a more comprehensive evaluation of the model's performance, we present the AUROC and F1-score results in this section. As shown in Table 5, overall, our method achieves the top average rank and the highest mean score across datasets. Compared to DTDG models, our method achieves the best performance on five out of the six datasets. In the case of AS-733, while the performance is constrained by the backbone Roland-GRU, PromptDyG yields a remarkable relative enhancement of 6.11% in F1-score. Compared with TTA methods, PromptDyG achieves the best performance on most datasets. Meanwhile, TTA methods also achieve more competitive results than the backbone across these metrics, indicating that aligning representations to structural shifts during the inference phase is crucial for accurate future link prediction.

Additionally, Table 6 reports the AUROC and F1-score of enabling existing DTDG methods with our PromptDyG. From Table 6, we can observe results consistent with the MRR reported in Table 2, with maximum improvements of 4.28% in AUROC and 6.79% in F1-score, indicating that our method comprehensively improves the backbone's performance through inference-time refinement.

## C. Extended Related Work Discussion

**Dynamic Graph Neural Networks.** Dynamic graph representation learning focuses on encoding temporal node embeddings for evolving graph structures. These methods are typically organized into two main categories—Continuous-Time Dynamic

*Table 5.* AUROC and F1-Score on six datasets. Best results are highlighted in bold and green, and second-best results are highlighted in purple. The Avg. Rank shows the average ranking of each method across all datasets, where rankings are computed independently for each dataset and then averaged. The last column is the average (AUROC/F1-Score) performance across all datasets.

| Metric | Methods | Datasets | | | | | | Avg. Rank | Avg. performance |
|---|---|---|---|---|---|---|---|---|---|
| | | AS-733 | Reddit-title | Reddit-body | UCI | Bitcoin-OTC | Bitcoin-Alpha | | |
| AUROC | | DTDG models | | | | | | | |
| | EvolveGCN-H | 0.7867 | 0.9258 | 0.8763 | 0.5630 | 0.5567 | 0.5710 | 13.00 | 0.7133 |
| | EvolveGCN-O | 0.6357 | 0.6428 | 0.5925 | 0.5340 | 0.5002 | 0.5038 | 14.00 | 0.5682 |
| | GCRN-GRU | 0.9819 | 0.9773 | 0.9608 | 0.8975 | 0.9227 | 0.9139 | 6.83 | 0.9424 |
| | GCRN-LSTM | **0.9825** | 0.9772 | 0.9606 | 0.8974 | 0.9176 | 0.9148 | 7.17 | 0.9417 |
| | GCRN-Baseline | 0.9821 | 0.9767 | 0.9588 | 0.9016 | 0.9216 | 0.9153 | 7.00 | 0.9427 |
| | TGCN | 0.9739 | 0.9760 | 0.9575 | 0.8313 | 0.8634 | 0.8771 | 10.83 | 0.9132 |
| | SFDyG | 0.9794 | 0.9809 | 0.9642 | 0.9053 | 0.8982 | 0.9012 | 6.50 | 0.9382 |
| | Roland-Average | 0.9570 | 0.9799 | 0.9693 | 0.8731 | 0.9317 | 0.9249 | 7.17 | 0.9393 |
| | Roland-MLP | 0.9208 | 0.9778 | 0.9485 | 0.8859 | 0.8631 | 0.8798 | 10.33 | 0.9127 |
| | Roland-GRU | 0.9732 | 0.9790 | 0.9707 | 0.9085 | 0.9381 | 0.9417 | 5.50 | 0.9519 |
| | | TTA models | | | | | | | |
| | +Tent | 0.9763 | 0.9531 | 0.9744 | **0.9189** | 0.9447 | 0.9456 | 4.67 | 0.9522 |
| | +Matcha | 0.9666 | 0.9762 | 0.9722 | 0.8696 | 0.9382 | 0.9426 | 7.17 | 0.9442 |
| | +GTrans | 0.9747 | 0.9824 | 0.9733 | 0.9183 | 0.9458 | **0.9473** | 3.00 | 0.9570 |
| | PromptDyG | 0.9779 | **0.9840** | **0.9776** | **0.9189** | **0.9464** | 0.9465 | 1.83 | 0.9586 |
| F1-Score | | DTDG models | | | | | | | |
| | EvolveGCN-H | 0.5115 | 0.7328 | 0.6237 | 0.3245 | 0.4715 | 0.4816 | 13.50 | 0.5243 |
| | EvolveGCN-O | 0.4169 | 0.5170 | 0.0451 | 0.4542 | 0.6418 | 0.5851 | 13.17 | 0.4434 |
| | GCRN-GRU | **0.9335** | 0.9289 | 0.9013 | 0.8104 | 0.8497 | 0.8374 | 6.50 | 0.8769 |
| | GCRN-LSTM | 0.9327 | 0.9284 | 0.9018 | 0.8089 | 0.8483 | 0.8426 | 7.00 | 0.8771 |
| | GCRN-Baseline | 0.9313 | 0.9277 | 0.8991 | 0.8128 | 0.8479 | 0.8459 | 7.33 | 0.8775 |
| | TGCN | 0.9212 | 0.9240 | 0.8928 | 0.7165 | 0.7631 | 0.7848 | 9.83 | 0.8337 |
| | SFDyG | 0.9114 | 0.9367 | 0.9058 | 0.7890 | 0.8548 | 0.8667 | 6.00 | 0.8774 |
| | Roland-Average | 0.8618 | 0.9332 | 0.9134 | 0.7879 | 0.8585 | 0.8558 | 7.00 | 0.8684 |
| | Roland-MLP | 0.6310 | 0.9186 | 0.6740 | 0.7014 | 0.5800 | 0.5710 | 12.17 | 0.6793 |
| | Roland-GRU | 0.8360 | 0.9288 | 0.9154 | 0.8000 | 0.8780 | 0.8747 | 6.67 | 0.8722 |
| | | TTA models | | | | | | | |
| | +Tent | 0.9225 | 0.9097 | 0.9243 | 0.8418 | 0.8941 | 0.8965 | 4.00 | 0.8982 |
| | +Matcha | 0.9089 | 0.9289 | 0.9211 | 0.7839 | 0.8809 | 0.8834 | 5.67 | 0.8845 |
| | +GTrans | 0.8919 | 0.9348 | 0.9226 | 0.8371 | 0.8919 | 0.8955 | 4.00 | 0.8956 |
| | PromptDyG | 0.8971 | **0.9401** | **0.9288** | 0.8420 | **0.8948** | **0.8976** | 2.17 | 0.9001 |

Graphs (CTDGs) and Discrete-Time Dynamic Graphs (DTDGs)—according to their temporal modeling strategies. CTDG-based methods treat dynamic graphs as event streams with accurate timestamps, generating dynamic node embeddings by iteratively processing information gathered from temporal neighbors (Kumar et al., 2019). Some approaches employ dynamic random walks to depict structural changes (Nguyen et al., 2018; Wang et al., 2021b), whereas others introduce specialized time encoders to embed temporal context into structural representations (Cong et al., 2023; Yu et al., 2023). Moreover, some researchers utilize temporal point processes to model structural evolution or node communications (Wen & Fang, 2022; Trivedi et al., 2019). Although those CTDG methods have demonstrated success, practical considerations such as privacy must be considered when collecting data in real-world scenarios, which makes it difficult to acquire fine-grained timestamps. By contrast, DTDGs (Yang et al., 2021; You et al., 2022; Qi et al., 2025) consist of a sequence of graph snapshots that describe all the events within specific time intervals, such as a day, a week, or a month. This snapshot-based formulation aligns better with the data collection mechanisms of many real-world systems, making DTDGs a practical choice for modeling long-term data evolution. Typically, the DTDG models (Pareja et al., 2020; Zhao et al., 2019; Seo et al., 2018; Hajiramezanali et al., 2019; Qi et al., 2025) combine GNNs with sequence models like RNNs to capture spatio-temporal dependencies, aiming to forecast the next graph snapshot. Despite sharing the goal of dynamic embedding, CTDG and DTDG methods differ significantly in data granularity and model design: CTDGs focus on micro-level event dynamics, whereas DTDGs prioritize macro-level structural evolution. This work primarily focuses on the critical issue that existing DTDG models overlook the test structural distribution shifts during the inference phase.

## D. Measuring Time Dependence via Neighborhood Similarity Dynamics

To characterize the structural evolution of dynamic graphs, we design a Temporal Neighborhood Change Rate (TNCR) to quantify the time dependence of neighborhood structures across consecutive snapshots.

**Jaccard Similarity Between Node Pairs.** Given a graph snapshot $G_t = (V, E_t)$ with adjacency matrix $\mathbf{A}_t$, we first compute the Jaccard similarity between the neighborhoods of all node pairs:

$$J_t(i,j) = \frac{|\mathcal{N}_t(i) \cap \mathcal{N}_t(j)|}{|\mathcal{N}_t(i) \cup \mathcal{N}_t(j)|}, \tag{14}$$

*Table 6.* AUROC and F1-Score results of enabling existing DTDG methods with our PromptDyG. Bold denotes better performance. Values in parentheses represent the percentage improvement over the backbone.

| Metric | Method | Datasets | | | | | |
|---|---|---|---|---|---|---|---|
| | | AS-733 | Reddit-title | Reddit-body | UCI | Bitcoin-OTC | Bitcoin-Alpha |
| AUROC | EvolveGCN-H | 0.7867 | 0.9258 | 0.8763 | 0.5630 | 0.5567 | 0.5710 |
| | +PromptDyG | **0.7936**(+1.58%) | **0.9396**(+1.38%) | **0.8869**(+1.06%) | **0.5720**(+0.90%) | **0.5995**(+4.28%) | **0.5835**(+1.25%) |
| | GCRN-GRU | 0.9819 | 0.9773 | 0.9608 | 0.8975 | 0.9227 | 0.9139 |
| | +PromptDyG | **0.9828**(+0.09%) | **0.9792**(+0.19%) | **0.9647**(+0.39%) | **0.9263**(+2.88%) | **0.9288**(+0.61%) | **0.9194**(+0.55%) |
| | GCRN-Baseline | 0.9821 | 0.9767 | 0.9588 | 0.9016 | 0.9216 | 0.9153 |
| | +PromptDyG | **0.9826**(+0.05%) | **0.9787**(+0.20%) | **0.9639**(+0.65%) | **0.9200**(+1.84%) | **0.9237**(+0.21%) | **0.9211**(+0.58%) |
| | TGCN | 0.9739 | 0.9760 | 0.9575 | 0.8313 | 0.8634 | 0.8771 |
| | +PromptDyG | **0.9762**(+0.23%) | **0.9792**(+0.32%) | **0.9640**(+0.65%) | **0.8480**(+1.67%) | **0.8793**(+1.59%) | **0.8838**(+0.67%) |
| | SFDyG | 0.9724 | 0.9809 | 0.9642 | 0.9013 | 0.8982 | 0.9012 |
| | +PromptDyG | **0.9773**(+0.49%) | **0.9816**(+0.07%) | **0.9688**(+0.42%) | **0.9042**(+0.29%) | **0.9056**(+0.74%) | **0.9043**(+0.31%) |
| F1-Score | EvolveGCN-H | 0.5115 | 0.7328 | **0.6237** | 0.3245 | 0.4715 | 0.4816 |
| | +PromptDyG | **0.5621**(+5.06%) | **0.7698**(+3.70%) | 0.6165(-0.72%) | **0.3712**(+4.67%) | **0.5389**(+6.79%) | **0.5204**(+3.88%) |
| | GCRN-GRU | 0.9335 | 0.9289 | 0.9013 | 0.8104 | 0.8497 | 0.8374 |
| | +PromptDyG | **0.9486**(+1.51%) | **0.9334**(+0.45%) | **0.9088**(+0.75%) | **0.8566**(+4.62%) | **0.8697**(+2.00%) | **0.8516**(+1.42%) |
| | GCRN-Baseline | 0.9313 | 0.9277 | 0.8991 | 0.8182 | 0.8479 | 0.8459 |
| | +PromptDyG | **0.9463**(+1.50%) | **0.9326**(+0.49%) | **0.9076**(+0.85%) | **0.8505**(+3.23%) | **0.8757**(+2.78%) | **0.8592**(+1.33%) |
| | TGCN | 0.9212 | 0.9240 | 0.8928 | 0.7165 | 0.7631 | 0.7848 |
| | +PromptDyG | **0.9251**(+0.39%) | **0.9322**(+0.82%) | **0.9054**(+1.26%) | **0.7634**(+4.69%) | **0.8019**(+3.88%) | **0.8092**(+1.91%) |
| | SFDyG | 0.9114 | 0.9367 | 0.9058 | 0.7890 | 0.8548 | 0.8667 |
| | +PromptDyG | **0.9151**(+0.37%) | **0.9381**(+0.14%) | **0.9145**(+0.87%) | **0.8321**(+4.31%) | **0.8649**(+1.01%) | **0.8705**(+0.38%) |

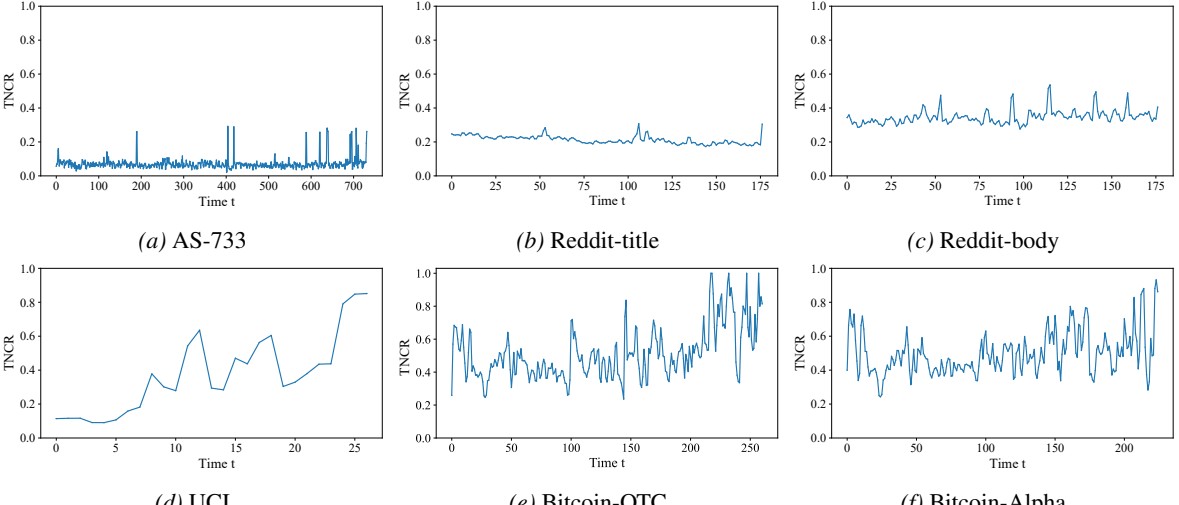

*(a)* AS-733     *(b)* Reddit-title     *(c)* Reddit-body

*(d)* UCI     *(e)* Bitcoin-OTC     *(f)* Bitcoin-Alpha

*Figure 11.* Temporal Neighborhood Change Rate (TNCR) curves on six datasets.

where $\mathcal{N}_t(i)$ denotes the neighbor set of node $i$ at time $t$.

**Temporal Difference of Neighborhood Similarity.** To capture the structural change between consecutive snapshots, we compute the absolute difference between Jaccard matrices:

$$D_t(i,j) = |J_{t+1}(i,j) - J_t(i,j)|. \tag{15}$$

This difference matrix highlights node pairs whose neighborhood similarity changes over time, effectively reflecting the local structural reorganization of the network.

**Global Temporal Neighborhood Change Rate.** We further define a global scalar to summarize the overall structural variation between two snapshots:

$$\Delta_t = \frac{1}{|\Omega_t|} \sum_{(i,j) \in \Omega_t} D_t(i,j), \tag{16}$$

where $\Omega_t$ is the set of non-zero entries in $D_t$. This metric measures the average Jaccard variation of all node pairs whose neighborhood relations have changed, which we term the Temporal Neighborhood Change Rate (TNCR).

A smaller $\Delta_t$ indicates that neighborhood structures at time $t + 1$ are highly consistent with those at time $t$, implying strong temporal dependence, while larger values reflect abrupt structural reorganization.

**Visualization of Temporal Structural Dynamics.** As shown in Figure 11, we visualize the TNCR sequence $\Delta_t$ of six real-world dynamic graphs as a time series curve, where the x-axis denotes snapshot index and the y-axis corresponds to the average difference $\Delta_t$. This visualization provides an intuitive overview of the structural changes over time. From Figure 11, we observe that AS-733, Reddit-title, and Reddit-body exhibit relatively small neighborhood change rates across consecutive snapshots, indicating strong temporal dependence where the network structures evolve smoothly over time. In contrast, UCI, Bitcoin-OTC, and Bitcoin-Alpha present consistently larger and more irregular variations, suggesting weak temporal dependence and highly volatile structural dynamics. This implies that their future topologies are less predictable from historical snapshots, posing greater challenges for temporal representation learning and forecasting in these datasets.

