# OpenReview forum: "PromptDyG: Test-Time Prompt Adaptation on Dynamic Graphs"
_ICML.cc/2026/Conference — ICML 2026 regular_

### Official Review · Reviewer_UgCh · 2026-03-05

**Soundness:** 2
**Presentation:** 3
**Significance:** 3
**Originality:** 3
**Overall Recommendation:** 4
**Confidence:** 3

**Summary:**

The paper proposes PromptDyG, a test-time prompt adaptation method for discrete-time dynamic graphs in the online/live-update setting. A pre-trained DTDG backbone is frozen, and a lightweight learnable prompt is optimized at each time step via unsupervised feature-wise entropy minimization to mitigate structural distribution shift between training and test snapshots. Experiments on 6 datasets show improved MRR compared to DTDG baselines and some representative TTA methods.

**Compliance With Llm Reviewing Policy:**

Affirmed.

**Final Justification:**

The second round response from the authors make me to raise score by offering extra theoretical supports, though the provided evidence is not yet a full justification.

**Key Questions For Authors:**

1. Please clarify what is observed at time $t$ during evaluation, and what $A^{\text{train}}_t/A^{\text{val}}_t/A^{\text{test}}_t$ mean versus test-period snapshots.

2. Please provide an ablation on the number of prompt optimization steps and report wall-clock time per snapshot, since test-time optimization is important for the paper's claim.

3. The theory part should avoid the dimension-as-prototype claim or reframing it as an assumption first in the main text (and validating empirically, e.g., with prototype vectors learned or extracted).

4. The Lemma in the appendix is overstated; this is rather a supporting intuition than a theoretical guarantee. (I) “converge toward its most relevant prototype” is not proved (the proof only provides a local gradient direction argument); no convergence analysis (no conditions like convexity, Lipschitz gradients, step-size bounds), no guarantee that it converges to a single prototype rather than just making $q_i$ more peaked, no discussion of stationary points (entropy minimization can have many). (II) ".. introduce $K$ evolutionary prototypes...." as in lines 614-616. What is $\mu_k$? learned vectors? basis? (III) The authors define entropy as $L_i = H(q_i)$ as in lines 620-621, but later in the gradient in line 637 / lines 653-654, the entropy becomes $H(p_i)$? (IV) The authors write the gradient using the Jacobian as in lines 663-665 and conclude the sharpening direction is “faithfully propagated” to $P_t$ as in lines 668-669. This is not guaranteed, as the Jacobian could be ill-conditioned, low-rank, or locally misaligned. Without standing conditions or assumptions on $J_i$, you can't claim the prompt update meaningfully realizes the embedding-level direction.

If questions and limitations are properly addressed during rebuttal, the reviewer will raise the score.

**Limitations:**

The limitations of this work and the impact statement are not included in the paper. The paper can be further improved by resolving the questions raised.

**Strengths And Weaknesses:**

Strengths

1. Clear problem definition and task setting: focuses on online dynamic graph prediction where structural shift is common (Fig. 1).

2. Simple intervention: it adapts only a prompt while keeping the backbone frozen (Fig. 3), which is practical for stability and deployment.

3. Comprehensive empirical results: Table 1 indicates PromptDyG ranks best across datasets and improves over strong dynamic baselines. Table 2 suggests it can work as a plug-in to DTDG backbones.

4. Helpful visuals: similarity distribution (Fig. 2) and t-SNE (Fig. 4) qualitatively align with the shift-mitigation claim.

Weaknesses

1. Notation ambiguity: "the test graph $A^{\text{test}}_t$" is input into the model to predict the subsequent snapshot” (as in lines 217–218). Later, $A^{\text{test}}$ is used to denote a test split within a snapshot (as in lines 299-300).

2. Theoretical framing is somewhat questionable: "each dimension $k$ ... can be viewed as a prototype $\mu_k$ " (as in lines 231–234). Treating embedding dimensions as prototypes is nonstandard and under-justified. The argument may not reflect how learned representations actually behave in practice.

3. Complexity claim possibly underestimates optimization cost: the paper states entropy loss cost is $O(SK)$ for each step (as in lines 297–300), but prompt adaptation performs multiple gradient steps per snapshot.

---

> ### Author Rebuttal · Authors · 2026-03-31
>
> We sincerely appreciate your constructive and positive comments on our design and empirical justification. Please see our response to your comments one by one below.
>
> > **Weaknesses #1 & Question #1** Notation ambiguity.
>
> Please see our response to **Reviewer hm2b Weaknesses #5**
>
> > **Weaknesses #2 & Question #3 & #4(II)** Treating embedding dimensions as prototypes is nonstandard.
>
> Please see our response to **Reviewer s1jP Weaknesses #1**.
>
> > **Weaknesses #3** The complexity claim underestimates optimization cost.
>
> We would like to clarify that, for each epoch, the computational complexity of the entropy loss is $O(SK)$; for prompt adaptation involving $e$ gradient steps (epochs) per snapshot, the cumulative complexity is $O(eSK)$.
>
> > **Question #2** An ablation on the optimization steps and running time.
>
> ```
> Table A1. MRR (running time/ms) across different epochs.
> ```
> |Epochs|1|5|10|
> |---|---|---|---|
> |Bitcoin-OTC|0.1998(13)|0.1944(67)|0.2004(146)|
> |Reddit-body|0.3906(73)|0.3873(319)|0.3879(582)|
>
> In Table A1, we conduct the ablation study of optimization steps (epochs) and runtime per snapshot. Optimal epochs vary across datasets: on Reddit-body, due to its strong temporal dependencies, we achieve best performance at epoch=1, while Bitcoin-OTC can benefit from more steps. Notably, performance remains stable across various epochs on both datasets, demonstrating our method's robustness to this hyperparameter. We also report the runtime (in parentheses), including both adaptation and inference, to highlight the good trade-off of efficiency and accuracy.
>
>
> > **Question #4 (III)** The definition of entropy.
>
> To ensure notation consistency and avoid redundant symbols, we'll correct the typo in the entropy term $H(p\_i) \to H(q\_i)$, and update $z\_{ik} \to \tilde{h}\_{ik}$ and $\tilde{h}\_i \to h\_i$ in Eq. (7).
>
> > **Question #4 (I)** The Lemma in the appendix has no convergence analysis.
>
> We'll add the following convergence analysis, including convexity, stationary point, Lipschitz gradients, and step-size bounds. For any node $i$, the distribution $q\_i = softmax(\tilde{h}\_i)$ maps to the probability simplex $\Delta^{K-1}$, which is a standard compact convex set. In information theory, the entropy loss function $\mathcal{L} = -\sum\_k q\_{ik} \log q\_{ik}$ in Eq.(8) is strictly concave (its second derivative is strictly negative definite), yielding a unique global maximum at the interior stationary point (i.e., the uniform distribution $q\_i^{\*}= [1/K, \dots, 1/K]^\top$). According to Jensen’s inequality and the representation theorem of compact convex set, the global minima of a strictly concave function ($\mathcal{L}$) exclusively lie at the extreme points of $\Delta^{K-1}$ (i.e., the $K$ one-hot vectors). Consequently, gradient descent for $\mathcal{L}$ pushes $q\_i$ to asymptotically approach a one-hot distribution (from high to low entropy), driving continuously maximizing the alignment $h\_i \to \mu\_{k^{\*}}$ with a specific prototype while suppressing others (i.e., $\tilde{h}\_{ik^{\*}}\uparrow，\tilde{h}\_{ij}(j\neq k^{\*})\downarrow$). In Eq.(9), $\lim\_{q\_{ik} \to 0} q\_{ik} \log q\_{ik} = 0$ and $H(q\_i)\geq0$ naturally bounds the gradient, and the softmax function acts as a strict 1-Lipschitz non-negative activation. The gradient mapped back to the real-valued latent space $\mathbb{R}^K$ remains globally bounded with a Lipschitz constant $L \approx 1$. Per the Descent Lemma, stable convergence is ensured when the step size $\eta \le 1/L$. Given $L \approx 1$, any standard learning rate $\eta \in (0, 1]$ (e.g., $10^{-3}$ in Adam) strictly satisfies this theoretical bound, guaranteeing monotonic decrease and preventing catastrophic oscillations during adaptation.
>
>
> > **Question #4 (IV)** Without standing assumptions on  $\mathbf{J}$.
>
>
> As suggested, we supplement a reasonable assumption: the input-output Jacobian $\mathbf{J}$ of the pre-trained backbone is non-degenerate.
>
> This assumption is justified by our empirical evidence. A properly pre-trained backbone can learn semantically meaningful node representations from input data; however, a low-rank Jacobian would imply feature collapse, which contradicts the model's observed efficacy. Consequently, a well-functioning pre-trained model provides a guarantee that its input-output Jacobian is neither ill-conditioned nor rank-deficient (non-degenerate). This is verified by t-SNE visualizations (Figs. 4 and 10), where the successful prompt-based realignment of latent representations confirms an effective gradient flow.
>
> > **Limitation #1** The limitations are not included.
>
> Please see our detailed response to **Reviewer Bqe2's Weakness #3**.

---

> > ### Author Rebuttal · Reviewer_UgCh · 2026-03-31
> >
> > The rebuttal does a solid job on the concrete engineering and clarity issues, but it does not fully resolve the theoretical issues: prototype story and convergence / Jacobian objections. Follow-up questions are listed below.
> >
> > 1. In the paper, the theory section says each latent dimension can be viewed as a prototype, then defines prototype scores using vectors $\mu_k$, which is already a bit internally messy. In rebuttal, the authors shift it to a different interpretation: the prototypes are columns of the final linear layer, and they give one empirical cosine-similarity check on Bitcoin-OTC. So, it is not a full justification of the original claim as written in the paper. That said, from my perspective, an interesting proposition + decent empirical evidence is way more convincing than a partially justified theory plus somewhat ad-hoc explanations.
> >
> > 2. The rebuttal shows that entropy over $q$ on the simplex has one-hot minima, then adds boundedness/Lipschitz language and a “non-degenerate Jacobian” assumption. But my concern was about the actual optimization of the prompt through the composed frozen-backbone model, not entropy as a standalone function of $q$. The rebuttal does not really prove convergence of the real prompt optimization problem, does not convincingly rule out problematic stationary points of the composed objective, and the Jacobian assumption is asserted from empirical behavior/t-SNE rather than established. So this still reads more like intuition than a convincing theorem, similar to point 1, the author could frame the narrative more persuasively by starting at a well-motivated proposition + supporting empirical evidence.
> >
> > Post-rebuttal:
> > Though the current evidence is not yet a proper justification, I would like to raise the score. Good luck.

---

> > > ### Author Response · Authors · 2026-04-02
> > >
> > > Please find our response to your follow-up question as follows.
> > >
> > > > **Follow-up question with Question #3** It is not a full justification of the original claim as written in the paper.
> > >
> > > Thanks for your comment. We would like to clarify that **our statement/claim remains the same:** the pre-trained model encodes $K$ different topologically evolutionary prototypes. We corrected the definition of the prototypes to avoid the use of the potentially misleading word 'dimension', which does not influence the validity of Lemma 4.1 (Pattern Refinement) and Proposition 4.2 (Margin Expansion) in the paper. We can obtain a consistent claim with Lemma 4.1: minimizing $\mathcal{L}\_{ent}$ via gradient descent on $\mathbf{P}\_t$ at time $t \in [1,T]$ forces the latent embedding $h\_i$ to converge toward its most relevant evolutionary prototype $\mu\_k$. This ensures that the inference embedding $\tilde{h}\_i$ achieves the maximal response toward the evolutionary prototype $\mu\_k$, thereby efficiently and continuously model the evolving patterns. Per Lemma 4.1, driven by the entropy minimization, positive node pairs tend to converge toward the same prototype, increasing their similarity while decreasing that of negative pairs. This effectively enlarges the similarity margin between positive and negative samples, consistent with Proposition 4.2.
> > >
> > > Furthermore, we add a proposition about prototypes and provide a theoretical foundation supporting it. Please see our detailed response to **Reviewer s1jP's follow-up weakness**.
> > >
> > > > **Follow-up question with Question #4(I)** The optimization of the composed objective and supporting theoretical evidence.
> > >
> > > We first provide a theoretical grounding for the non-degeneracy of the Jacobian $\mathbf{J}$. Particularly, the input-output Jacobian of the backbone is defined as $\mathbf{J} = \frac{\partial \tilde{H}}{\partial P}$, where $\tilde{H} = f\_\theta(P) \in \mathbb{R}^{N \times K}$ is the final output and $P$ is learnable prompt input. To facilitate a rigorous analysis within the framework of linear operators, we employ the vectorization operator $\text{vec}(\cdot)$ to recast $\mathbf{J}$ into a standard matrix form: $\mathbf{J} = \frac{\partial \text{vec}(\tilde{H})}{\partial \text{vec}(P)} \in \mathbb{R}^{NK \times ND}$.  Then since the prompt parameters provide $ND$ degrees of freedom for a $NK$-dimensional output, $\mathbf{J}$ is a wide matrix ($ND > NK$) whenever $D > K$ (see Table A1 below). Leveraging over-parameterization theory and Neural Tangent Kernel theory [1,2], such over-parameterized regimes ensure that the Gram matrix $\mathbf{J}\mathbf{J}^\top$ remains strictly positive-definite during optimization. Consequently, the full-row-rank property of $\mathbf{J}$ is structurally guaranteed, ensuring that gradient signals are effectively back-propagated to the prompt space.
> > >
> > > ```
> > > Table A1. The dimensions $D$ and $K$ on six datasets.
> > > ```
> > > ||AS-733|Reddit-title|Reddit-body|UCI|Bitcoin-OTC|Bitcoin-Alpha|
> > > |---|---|---|---|---|---|---|
> > > |$D$|200|300|300|200|200|200|
> > > |$K$|128|128|64|64|64|64|
> > >
> > > Secondly, we perform a convergence analysis on the prompt-based composed objective $\mathcal{J}(P) = \mathcal{L}(f\_\theta(P))$. Following the chain rule, we can derive $\nabla\_{P} \mathcal{J} = \mathbf{J}^\top \nabla_{\tilde{H}} \mathcal{L}$. Under the non-degenerate Jacobian assumption, the total gradient is $\nabla\_{P} \mathcal{J}=0$ if and only if $\nabla_{\tilde{H}} \mathcal{L}=0$. This ensures that the optimization avoids problematic stationary points or gradient vanishing before reaching the one-hot distribution. Furthermore, the backbone $f_\theta$ is a composition of bounded-spectral-norm operators (e.g., linear, 1-Lipschitz ReLU layers). This ensures $f_\theta$ is Lipschitz continuous with a bounded Jacobian $||\mathbf{J}|| \le L\_f$. Since the entropy gradient $\nabla \mathcal{L}$ is also locally Lipschitz due to the Softmax-smoothed latent space, the composed gradient $\nabla_{P} \mathcal{J}$ is Lipschitz continuous with constant $L\_c$. By the Descent Lemma [3], having a step size of $\eta \le 1/L\_c$ guarantees a monotonic decrease in $\mathcal{J}(P)$.
> > >
> > > In addition, as the maximum-entropy state acts as an instability reducer, gradient descent drives the optimization toward the stable boundaries of the probability simplex, refining $P$ to align node embeddings with evolutionary prototypes.
> > >
> > > In our final version, we'll add the above theoretical foundations to support our proposition and model design.
> > >
> > > [1] Du, S. S., et al. (2019). Gradient descent provably optimizes over-parameterized neural networks. ICLR
> > >
> > > [2] Jacot, A., et al. (2018). Neural tangent kernel: Convergence and generalization in neural networks. NeurIPS.
> > >
> > > [3] Nesterov, Y. (2013). Introductory lectures on convex optimization: A basic course (Vol. 87). Springer Science and Business Media.
> > >
> > > Post-rebuttal: Thank you very much for the thoughtful comments and for raising the score.

---

### Official Review · Reviewer_s1jP · 2026-03-05

**Soundness:** 3
**Presentation:** 3
**Significance:** 3
**Originality:** 3
**Overall Recommendation:** 4
**Confidence:** 3

**Summary:**

This paper addresses structural distribution shift in Discrete-time Dynamic Graph learning by proposing PromptDyG, a test-time adaptation framework. It injects a lightweight learnable prompt into the input features while freezing the pre-trained backbone, and optimizes the prompt via unsupervised feature-wise entropy minimization to align representations with evolutionary patterns. experiments on datasets demonstrate consistent improvements over various DTDG backbones.

**Compliance With Llm Reviewing Policy:**

Affirmed.

**Final Justification:**

The rebuttal clarified several practical concerns, including the prompt reinitialization strategy, hyperparameter selection, and backbone sensitivity, and the additional experiments on anomaly detection broaden the empirical scope. Overall, I maintain my current score.

**Key Questions For Authors:**

1. What size prompt is used in the main experiment?While the ablation experiments demonstrate that the model is robust to Prompt size, in the comparison experiments in the main tables 1 and 2. If node-level Prompts (N × D) are being used, a large number of parameters are introduced into the large-scale graph, which is contrary to the lightweight.
2. promptDyG uses the live-update evaluation setting, but emphasizes that the backbone is frozen in the inference phase. I am a little puzzled about this, whether it is frozen, and if the live only updates the prompt, whether the live-update in the upper left corner of Figure 3 is easy to cause misunderstandings
3. The use of the Adam optimizer with learning rates chosen from {0.01, 0.001, 0.005, 0.0005}, the model is not allowed to touch the test set labels for hyperparameter search. Could the authors please clarify: how are these hyperparameters determined?

**Limitations:**

No, sensitivity to a pre-trained backbone network: the framework relies on a frozen backbone. Authors should discuss how the quality or bias of the initial pre-training affects TTA performance. If the backbone is poorly trained or biased, does rapid adaptation amplify these problems?

**Strengths And Weaknesses:**

## Strengths:
1. Interesting motivation: Combining Test-Time Adaptation with prompt learning for dynamic graphs is a novel idea. Most existing TTA works focus on static graphs, CV or other tasks
2. Strong empirical results: The performance gain on several datasets seems consistent and significant compared to original backbones and other TTA baselines.
3. Good visualization: The authors provide t-SNE plots and similarity score distribution curves to visualize how the prompt helps to bridge the distribution gap.
## Weaknesses:
1. Strong theoretical assumptions: The authors assume that different prototypes are quasi-orthogonal in a well-trained backbone. However, Backbone simply pretrains for link prediction using a regular BCE , and this loss function does not guarantee the formation of mutual orthogonality in the hidden space.
2. Implementation details unclear: The authors provide detailed implementation details, but many of the details are dynamic, such as the selected learning rate. This can create concern that it is not clear on what basis the choice was made, and if it is based on test results, then the experiment is not justified.

---

> ### Author Rebuttal · Authors · 2026-03-31
>
> Thanks for the constructive suggestions and the positive comments.
>
> > **Weaknesses #1** Quasi-orthogonal assumption of prototypes.
>
> Our method is built upon the assumption that the pre-trained latent space encodes $K$ diverse evolutionary prototypes, captured by the learned weight matrix $W = [\mu\_1, \dots, \mu\_K] \in \mathbb{R}^{d \times K}$ in the final linear layer.  To empirically justify this assumption, we calculate pairwise cosine similarities of the $K$ prototypes on Bitcoin-OTC. Excluding the diagonal, the similarity matrix yields a near-zero mean of $0.003 \pm 0.08$. These exceptionally low values signify that the $K$ prototypes are highly dissimilar and well-separated, effectively precluding mode collapse and capturing distinct evolutionary patterns. We'll also add results on the other datasets.
>
> The inner product $h_i^\top \mu_k$ reflects the semantic matching degree between node $i$ and prototype $k$. Specifically, let $(u, v)$ be a positive pair that shares the same evolutionary pattern, then according to Lemma 1, driven by entropy minimization, the positive pair $(u, v)$ align towards the same prototype $\mu_c$ by maximizing their similarity: $s\_{uv} \to ||\mu\_c||^2$. Conversely, entropy minimization forces a negative pair $(u, w)$ to align with different prototypes, $\mu\_{c\_1}$ and $\mu\_{c\_2}$ ($c\_1 \neq c\_2$), and their similarity: $s\_{uw} \to \mu\_{c\_1}^\top \mu\_{c\_2}$. By the Cauchy-Schwarz inequality and the separation of distinct prototypes, their cross-similarity is strictly bounded: $\mu\_{c\_1}^\top \mu\_{c\_2} \ll ||\mu\_c||^2$, leading to an increase in the prediction margin, i.e., $\gamma=s\_{uv}-s\_{uw}>0$, which is also empirically justified in Fig. 2 and 9.
>
>
>
> > **Weaknesses #2 & Question #3** How are the hyperparameters determined?
>
> For the pre-training, the learning rate is determined based on the validation set, while the learning rate for the prompt adaptation is solely based on the unsupervised entropy loss of Eq. (4), where no test labels are involved.
>
> > **Question #1** What size prompt is used in the main experiment?
>
> While we report the performance of PromptDyG using an $(N \times D)$ prompt in Tables 1 and 2, Table A1 below shows that even a minimal prompt size (e.g., $1 \times D$) can still achieve state-of-the-art results. Therefore, for large-scale graphs, smaller prompt sizes can be flexibly adopted to ensure a lightweight nature with negligible variation in performance. Moreover, Figs. 8 and 11 confirm the high computational efficiency of our PromptDyG on all benchmarks.
>
> ```
> Table A1. MRR results of PromptDyG with different prompt sizes and the best-performing baseline (GTrans).
> ```
> |Methods|Reddit-title|Reddit-body|
> |---|---|---|
> |GTrans |0.4276|0.3712|
> |PromptDyG($1 \times D$)|0.4406|0.3902|
> |PromptDyG |0.4425|0.3906|
>
> > **Question #2** Confusions on the live-update evaluation protocol and Figure 3.
>
> For detailed live-update setting, please refer to the response to **Reviewer hm2b Weaknesses #1 and #5**. We will reorganize Fig. 3 to make the whole process clearer.
>
>
> > **Limitation #1** Sensitivity to the quality of the pretrained backbone.
>
> ```
> Table A2. MRR results using backbones of different quality.
> ```
> |Methods|Reddit-body|Reddit-body|UCI|UCI|
> |---|---|---|---|---|
> |---|t=1|t=10|t=1|t=10|
> |Roland-GRU|0.1732|0.2592|0.0841|0.1082|
> |PromptDyG|0.2173|0.2748|0.1079|0.1293|
>
> We conduct an additional experiment in which the backbone (Roland-GRU) is trained only using a small number of $t$ time steps ($t=1$ or $10$) and remains frozen thereafter, resulting in a deliberately under-fitted backbone. As shown in Table A2, even when the backbone performance degrades substantially, PromptDyG with TTA still yields significant improvements. This confirms that the performance gains from TTA remain consistently positive, even under unfavorable backbone conditions. The underlying reason is that, despite being under-fitted, the trained backbone still captures certain evolutionary patterns, on which our TTA method can still effectively align the shifted test data with the captured evolutionary patterns, gradually calibrating the uncertain predictions for positive and negative pairs, thereby achieving robust and consistent performance gains.
>
> Moreover, the visualizations (Figs. 2, 4, 9, and 10) also verify that TTA effectively alleviates test-time distribution shifts.

---

> > ### Author Rebuttal · Reviewer_s1jP · 2026-04-01
> >
> > The rebuttal improves the clarity of several practical aspects, but the theoretical foundations still rely on assumptions that are empirically motivated rather than formally established. I maintain my current positive score.

---

> > > ### Author Response · Authors · 2026-04-02
> > >
> > > We greatly appreciate your further comments, and it is great to know that our response has addressed some of your questions. Please find our response to your remaining concerns below.
> > >
> > > > **Follow-up weakness with Weakness #1** The theoretical foundations still rely on assumptions that are empirically motivated rather than formally established.
> > >
> > > We'll add the following proposition about the prototypes. Let $H\in\mathbb{R}^{N \times d}$ denote the node representations learned by the backbone and $\tilde{H} = H W$ be the final embedding, where $W =[\mu\_1,\mu\_2,\cdots,\mu\_K]\in \mathbb{R}^{d \times K}$ is a learnable projection matrix. Under the link prediction objective in Eq.(2), the resulting representation space lies in a $K$-dimensional subspace spanned by the columns of $W$, where each column vector $\mu\_k$ can be interpreted as a structural prototype.
> > >
> > > Please note that $\mu\_k$ is the column vector of the projection matrix $W$ learned by pre-trained backbones, and these vectors do not necessarily have orthogonality among them. Moreover, whether these prototype vectors are orthogonal or not will not affect the establishment of this proposition, nor does it influence the claim of Lemma 4.1 and Proposition 4.2 in the paper. Regarding the validity of Lemma 4.1 and Proposition 4.2 under this proposition, please refer to our response to **Reviewer UgCh's follow-up question with Question #3**.
> > >
> > > Meanwhile, we provide a theoretical foundation supporting this proposition as follows. Since $\tilde{H} = H W$, each node embedding satisfies $\tilde{h}\_i=h\_iW$, which implies that every embedding is a linear combination of the column vectors of $W$. Let $W = [\mu\_1,\mu\_2,\cdots,\mu\_K]$, then $\tilde{h}\_i=\sum\_{k=1}^K\alpha\_{ik}\mu\_k$ for some coefficients $\alpha\_{ik}$, indicating that all embeddings lie in the subspace $\text{span}(W) =\text{span}(\mu\_1,\mu\_2,\cdots,\mu\_K)$. Given the link probability between nodes $i$ and $j$ defined as $\sigma(\tilde{h}\_i^\top\tilde{h}\_j)$, substituting $\tilde{H} = H W$ yields $\sigma(h\_i WW^\top h\_j)$ or in matrix form $\sigma(H WW^\top H^\top)$. Minimizing the link prediction loss is therefore equivalent to fitting $A \approx \sigma(H WW^\top H^\top)$, which can be interpreted as reconstructing the adjacency matrix.  Furthermore, let $M = WW^\top$, since $W \in \mathbb{R}^{d \times K}$, we have $\text{rank}(M)\leq K$, leading to a low-rank interaction model $A \approx \sigma(H M H^\top)$. Such low-rank graph models capture a finite number of dominant interaction patterns [1,2]. Accordingly, the embedding space is confined to the subspace spanned by $\\{\mu\_1,\mu\_2,\cdots,\mu\_K\\}$, where each column vector $\mu\_k$ represents one prototypical direction describing a characteristic interaction pattern, and so the node embedding $\tilde{h}\_i=h\_iW$ can be interpreted as the projection of node $i$ onto these prototype directions. Therefore, the latent representation space admits a set of structural prototypes corresponding to the columns of $W$.
> > >
> > > [1] Holland, P. W., et al. (1983). Stochastic blockmodels: First steps. Social networks, 5(2), 109-137.
> > >
> > > [2] Hoff, P. D., et al. (2002). Latent space approaches to social network analysis. Journal of the american Statistical association, 97(460), 1090-1098.

---

### Official Review · Reviewer_Bqe2 · 2026-03-11

**Soundness:** 3
**Presentation:** 3
**Significance:** 3
**Originality:** 3
**Overall Recommendation:** 5
**Confidence:** 4

**Summary:**

This paper proposes PromptDyG, a novel discrete-time dynamic graph learning model that leverages unsupervised test-time prompt adaptation under a live-update online setting to address the distribution shift between training and test snapshots. By introducing a lightweight learnable prompt on a frozen backbone and optimizing it via feature-wise, label-free entropy minimization, PromptDyG adapts to structural shift without retraining the entire architecture. Both theoretical analysis and empirical results across six datasets demonstrate the effectiveness of this approach.

**Compliance With Llm Reviewing Policy:**

Affirmed.

**Final Justification:**

The authors’ response has addressed my concerns well, and their thorough rebuttal has further strengthened the paper. After reviewing the response, I would like to raise my score to 5.

**Key Questions For Authors:**

1, What would be the potential impact on model performance if the prompts are instead updated cumulatively over time, rather than being reinitialized for each snapshot?
2, Since dynamic graphs consist of CTDGs and DTDGs, can the proposed method be applied for CTDGs?

**Strengths And Weaknesses:**

Strengths:
1,The paper is generally well-written and easy-to-follow.
2,The paper addresses an important yet underexplored problem in DTDG and proposes a novel prompt-based adaptation framework to tackle structural distribution shifts in dynamic graphs.
3,The theoretical analysis and multiple visualization experiments (e.g., similarity margin distributions, t-SNE) provide intuitive and convincing evidence for the effectiveness of the proposed method.
4,The comprehensive experiments demonstrate the effectiveness and efficiency of the proposed method.
Weekness:
1, Are the prompts reinitialized for each snapshot? A clearer formal description of the adaptation schedule would improve reproducibility.
2, Some figures are not very clear, such as Figure 4. The authors should make them more readable.
3, A discussion of the limitations of the proposed approach is missing, which would help readers better understand potential challenges and future directions.
4, Please ensure consistency in the use of symbols, for example, the number of nodes N and n.

---

> ### Author Rebuttal · Authors · 2026-03-31
>
> We sincerely thank the reviewer for the positive and encouraging comments on the writing, novelty, and empirical justification. Please see our response to your comments one by one below.
>
>
> > **Weaknesses #1 & Question #1**  Are the prompts reinitialized for each snapshot?  What is the model performance when prompts are updated cumulatively?
>
> Thanks for your comment. In PromptDyG, prompts are reinitialized at each time step. Table A1 compares PromptDyG with PromptDyG-V1 (which uses cumulative updates).
> ```
> Table A1. MRR performance on two datasets.
> ```
> |Methods|Bitcoin-Alpha|Reddit-body |
> |:--- | :--- | :--- |
> |Roland-GRU|0.1547 |0.3622|
> |PromptDyG-V1|0.1679|0.3860|
> |PromptDyG|0.1732|0.3906|
>
> The superior performance of both prompt strategies over backbone Roland-GRU suggests that prompt-based adaptive learning effectively mitigates the uncertainty prediction induced by test-time distribution shifts. Secondly, PromptDyG outperforms PromptDyG-V1, indicating cumulative updates may introduce historical biases that hinder adaptation to the current test snapshot. In contrast, reinitialization enables more effective adaptation to instantaneous distribution shifts.
>
> > **Weaknesses #2** Some figures are not very clear.
>
> Thanks for your comment. We will enhance Fig. 4 by enlarging markers and improving the color contrast to show the two distributions more clearly.
>
> > **Weaknesses #3** Discussion of the limitations of the proposed approach.
>
> Thanks for your comment. A limitation of this work is that it focuses solely on the link prediction task in DTDGs. Future work will explore other data types, such as CTDGs, and extend our framework to various tasks. In response to **Reviewer hm2b Weaknesses #3**, we have added results for another task, dynamic graph anomaly detection, which will be added in our final version
>
> > **Weaknesses #4** Inconsistent use of symbols.
>
> Thank you for pointing out this. We will unify the notation for the number of nodes (using $N$ consistently).
>
> > **Question #2** Can the proposed method be applied to CTDGs?
>
> We appreciate the suggestion regarding CTDGs. Typically, CTDGs treat dynamic graphs as continuous event streams, denoted as $G = (V, E, T)$, where link prediction at time $t$ relies on historical interactions $\{(v'\_i, v'\_j, t') \mid t' < t\}$. In a standard CTDG evaluation, the model performs link prediction on a per-event basis. This granular inference lacks a structured data window or snapshot, making it difficult to characterize the distributional shift required for test-time adaptation methods like PromptDyG. In future, we consider anchor-event subgraph extraction to construct localized topological snapshots from the event stream, enabling our model to adapt to fine-grained evolutionary patterns for CTDGs.

---

> > ### Author Rebuttal · Reviewer_Bqe2 · 2026-04-03
> >
> > The authors’ response has addressed my concerns well, and their thorough rebuttal has further strengthened the paper. After reviewing the response, I would like to raise my score to 5.

---

> > > ### Author Response · Authors · 2026-04-06
> > >
> > > We're very pleased that our response has addressed your concerns. Thank you very much for the positive comments and for raising the recommendation score!

---

### Official Review · Reviewer_hm2b · 2026-03-13

**Soundness:** 2
**Presentation:** 2
**Significance:** 2
**Originality:** 3
**Overall Recommendation:** 4
**Confidence:** 4

**Summary:**

This paper studies test-time adaptation for DTDGs, focusing on temporal link prediction under a live-update setting. The key motivation is that existing dynamic graph methods are typically trained on historical snapshots and directly deployed on later snapshots, making them vulnerable to structural distribution shifts at inference time. To address this, the paper proposes PromptDyG, which freezes a pretrained DGNN backbone and learns a dynamic graph prompt at each test-time step via unsupervised feature-wise entropy minimization. The paper also provides a theoretical analysis to explain why entropy minimization can refine latent evolutionary patterns and enlarge the margin between positive and negative node pairs, and evaluates the method on six dynamic graph benchmarks against DTDG and test-time adaptation baselines.

**Compliance With Llm Reviewing Policy:**

Affirmed.

**Final Justification:**

The rebuttal addressed your main concerns

**Key Questions For Authors:**

Is it reinitialized at every step, or warm-started from P_{t-1}? This affects both the interpretation of the method and its practical efficiency.

**Limitations:**

Yes

**Strengths And Weaknesses:**

Strengths

1. Test-time adaptation for dynamic graphs is a meaningful and relatively underexplored problem. The paper identifies structural shift across time as a practical challenge for temporal link prediction.

2. The experiments cover six datasets and compare against multiple dynamic graph and TTA baselines. For temporal link prediction, the empirical study is reasonably comprehensive and the results validate the method’s effectiveness.

3. The theoretical analysis offers a coherent explanation of why entropy minimization may sharpen latent pattern assignments and improve discriminability. While not a strong proof, it helps justify the method.

Weaknesses

1. The task setting and evaluation protocol are not sufficiently clear, and their realism is not fully convincing
The paper seems to assume the following setting: given $A_t^{train}$ and $A_t^{test}$ at each time step, the DGNN backbone is first trained using the training sets from all time steps, and then the backbone is frozen while graph prompts are optimized on the test sets. This setting is questionable in terms of realism. In a more natural real-world scenario, the model should only have access to graph snapshots from times 1,…,t, in order to predict edges at time >t, rather than having access to data beyond the training horizon in advance. I believe a more natural and meaningful setting would be to train a DGNN to predict links at time t+1 based on historical snapshots 1,…,t; the truly important challenge at test time is then how the model handles distribution shift once the time steps go beyond the maximum training horizon $t_{\max}$.

2. The theoretical analysis is supportive rather than conclusive. The theory explains why entropy minimization may help under specific assumptions, such as latent dimensions behaving as evolutionary prototypes. However, it does not establish that prompt optimization will reliably converge to a good solution or remain robust more generally.

3. The empirical scope is somewhat narrow relative to the paper’s broader framing. The experiments focus almost entirely on temporal link prediction. If the goal is to present PromptDyG as a general framework as expressed in the paper title for dynamic graph test-time adaptation, broader task coverage would strengthen the claim.

4. Regarding the method design, while the method shows empirical gains, its core adaptation objective mainly forces uncertain outputs to become more confident through entropy minimization. This does not necessarily imply that the predictions become more correct; instead, it may simply sharpen incorrect beliefs. As a result, the method may worsen overconfidence, while also introducing new concerns regarding uncertainty reliability, calibration, and interpretability.

5. The presentation significantly increases the difficulty of understanding the paper, mainly in the following aspects:

  - The training protocol of the DGNN backbone remains confusing. According to the description around Eq. (2), the backbone is trained by predicting edges at the next time step, but the paper does not clearly specify the temporal boundary of this training process. As a result, it is difficult for the reader to understand the exact training range of the backbone and how it relates to the test phase.

- The relationship between A_t^{train} and A_t^{test} is unclear. From the way Figure 1 is presented, they do not seem to belong to the same graph; however, according to the textual description around Eq. (1), they appear to be different subsets within the same graph. This inconsistency substantially increases the difficulty of understanding the problem setting.

- The key problem setup is not explained clearly enough. For example, the paper does not clearly and consistently explain how the DGNN backbone is trained, where the distribution shift occurs, why it occurs, and what information is observable at each stage. Because these key points are not presented in a unified and clear way, the logical connection among the motivation, method, and experiments is not as smooth as it should be, which hurts the overall readability.

---

> ### Author Rebuttal · Authors · 2026-03-31
>
> Thanks for the constructive suggestions and the positive comments.
>
> > **Weaknesses #1** Concerns about the task setting and evaluation protocol.
>
> Dynamic graph learning is divided into the offline setting and the online setting (Fig. 1).
> In the offline setting, the model is trained using all historical snapshots ($1, ..., t$) and then evaluated on subsequent snapshots ($>t$) **without updates**, meaning the model is not adapted over time. In real applications, patterns are evolving with new data, on which dynamic models are needed to make adaptive future predictions [1]. Therefore, we adopt the live-update online setting following [1], where the model is updated at each time step to fit evolving data, mimicking real-world use cases. Specifically, at each time step $t$, the model is optimized to predict links at $t+1$ ($A_{t+1}^{test}$) based on all the observed data up to time $t$ and partial edges at $t+1$, i.e., $A_{t+1}^{train}$. After that, the model is updated on $A_{t+2}^{train}$ to perform the next round of prediction of $A_{t+2}^{test}$ in a rolling manner. At each time $t$, there are no overlaps between $A_{t}^{train}$ and $A_{t}^{test}$.
>
> For the online setting, it's worth exploring the suggested setting, i.e., without relying on $A_{t+1}^{train}$ to predict $t+1$. We leave it as an important direction for future work. Currently, to ensure a fair comparison with baselines, we adopt the mainstream online setting.
>
> PromptDyG can also be used under the offline setting, which is demonstrated by the below comparison with Roland-GRU in Table A1.
> ```
> Table A1. Results on Bitcoin-OTC under the offline setting.
> ```
> ||MRR|AUC|
> |:-|:-|:-|
> |Roland-GRU|0.1804|0.8882|
> |PromptDyG|0.1911|0.9159|
>
> [1] You, J., et al. Roland: graph learning framework for dynamic graphs. SIGKDD, 2022.
>
> > **Weaknesses #2** It does not establish that prompt optimization will reliably converge to a good solution.
>
> In the response to **Reviewer UgCh-Question #4(I)**, we add a convergence analysis. It demonstrates that the entropy loss drives the node distribution to asymptotically converge toward a one-hot distribution. This distribution implies that node representation aligns with a specific evolutionary prototype encoded in the pre-trained model.
>
> > **Weaknesses #3** The empirical scope is somewhat narrow.
>
> We further evaluate PromptDyG on dynamic graph anomaly detection (GAD). Following DP-DGAD [2], we pre-train on Bitcoin-Alpha to store normal and abnormal prototypes and adapt to Bitcoin-OTC via self-supervised alignment. PromptDyG is integrated into DP-DGAD through TTA to enable dynamic GAD. In Table A2, PromptDyG consistently improves AUC under two anomaly ratios, showing its effectiveness in mitigating the distribution shifts in dynamic GAD.
> ```
> Table A2. AUC on Bitcoin-OTC under two anomaly ratios.
> ```
> |Methods|1\%|5\%|
> |:-|:-|:-|
> |DP-DGAD|51.55|52.94|
> |+PromptDyG|54.03|56.20|
>
> [2] Zheng, J., et al. DP-DGAD: A generalist dynamic graph anomaly detector with dynamic prototypes, WWW, 2026.
>
> > **Weaknesses #4** Is the adaptation objective really reliable?
>
> We clarify that entropy minimization serves as an adaptation mechanism to mitigate predictive uncertainty from test-time distribution shifts. During inference, the structural divergence often causes output shifts (Figs. 4 and 10) and ambiguous predictions (Figs. 2 and 9). Our learnable prompts adapt to the shift by aligning test distributions with pre-trained evolutionary prototypes (Figs. 4 and 10), thereby widening the similarity margin between positive and negative pairs (Figs. 2 and 9) for more precise predictions. To verify this claim, an empirical justification is added, in our response to **Reviewer s1jP Limitation 1**, which shows that PromptDyG can achieve stable performance under varying contexts of the adaptation, i.e., using trained backbones of varying quality.
>
> > **Weaknesses #5** Clarification on training protocol, relationship between $A_t^{train}$ and $A_t^{test}$ and key problem setup.
>
> We will revise the paper to clarify the training protocol. Concretely, at each time step $t$, the DGNN backbone is trained using $A_t^{train}$ as input and $A_{t+1}^{train}$ as the supervision signal. Then, the backbone is frozen, with only the prompt being optimized over $A_t^{test}$ via the unsupervised entropy minimization. Finally, the learned prompt is applied to $A_t^{test}$ to predict $A_{t+1}^{test}$.
>
> The model is trained on $A_{t}^{train}$ but makes inference on $A_{t}^{test}$ to predict $A_{t+1}^{test}$ at each $t$, so there exist distribution shifts between training and inference. Note that $A_t^{train}$ and $A_t^{test}$ are indeed different subsets of $G_t$, rather than belonging to different snapshots. We apologize that Fig. 1 may have caused confusion and will reorganize it to ensure consistency with the textual description.
>
> > **Question #1** Is the prompt reinitialized or warm-started?
>
> Please refer to the response to **Reviewer Bqe2 Weaknesses #1**.

---

> > ### Author Rebuttal · Reviewer_hm2b · 2026-04-04
> >
> > Thanks for the rebuttal. I have raised my rating.

---

> > > ### Author Response · Authors · 2026-04-06
> > >
> > > Thank you very much for your consideration of our replies and for the raised recommendation score. We're very pleased to know that our response has addressed your concerns.

---

### Decision · Program_Chairs · 2026-04-30

**Decision:**

Accept (regular)

**Comment:**

This paper tackles link prediction problem from dynamic snapshots of graphs. Existing methods do not perform any update the model beyond a certain timestamp: as a result, they fail to capture the evolving changes across graphs and adapt to the shift in distributions. They provide a new method, where they freeze the backbone of the model and maximizing a label free entropy term with respect to test time prompt injected at feature.

The reviewers are generally positive. At a high level, a  key concern which I found common across the reviewers is about evaluation protocol: this involves hyperparameter search, what is available during training, validation and test (because test also involves optimization now). The authors should clarify this in the final revision.